# Dimerization-dependent serine protease activity of FAM111A prevents replication fork stalling at topoisomerase 1 cleavage complexes

Sowmiya Palani[1,6], Yuka Machida[2,3,6], Julia R. Alvey [4], Vandana Mishra[3], Allison L. Welter [1,3], Gaofeng Cui [4], Benoît Bragantini [4], Maria Victoria Botuyan [4], Anh T. Q. Cong [4], Georges Mer [4], Matthew J. Schellenberg [4,7] ✉ & Yuichi J. Machida [2,3,5,7] ✉

FAM111A, a serine protease, plays roles in DNA replication and antiviral defense. Missense mutations in the catalytic domain cause hyper-autocleavage and are associated with genetic disorders with developmental defects. Despite the enzyme's biological significance, the molecular architecture of the FAM111A serine protease domain (SPD) is unknown. Here, we show that FAM111A is a dimerization-dependent protease containing a narrow, recessed active site that cleaves substrates with a chymotrypsin-like specificity. X-ray crystal structures and mutagenesis studies reveal that FAM111A dimerizes via the N-terminal helix within the SPD. This dimerization induces an activation cascade from the dimerization sensor loop to the oxyanion hole through disorder-to-order transitions. Dimerization is essential for proteolytic activity in vitro and for facilitating DNA replication at DNA-protein crosslink obstacles in cells, while it is dispensable for auto-cleavage. These findings underscore the role of dimerization in FAM111A's function and highlight the distinction in its dimerization dependency between substrate cleavage and autocleavage.

FAM111A (FAM111 trypsin-like peptidase A) is a serine protease involved in multiple cellular processes including antiviral defense and DNA replication. The antiviral function of FAM111A was suggested in studies that identified FAM111A as a host restriction factor for the host range mutants of simian virus 40 (SV40) and orthopoxviruses[1–3]. In a subsequent proteomics study FAM111A was also found to localize at nascent DNA and was shown to promote DNA replication via interactions with proliferating cell nuclear antigen (PCNA) through its PCNA-interacting peptide (PIP) box[4]. Proper regulation of FAM111A appears to be crucial, given that heterozygous mutations in *FAM111A* cause two rare human syndromes, Kenny-Caffey Syndrome type 2 (KCS2) and the more severe disorder, Gracile Bone Dysplasia (GCLEB)[5–8], which are characterized by skeletal abnormalities, hypoparathyroidism, hypocalcemia, and low stature. Most of the disease-associated mutations are missense mutations clustered either within or around the peptidase domain of FAM111A[9]. These mutations are thought to cause hyperactivation of the enzyme as inferred from its hyper-autocleavage activity[10,11]. Ectopic expression of the disease-associated mutations causes impaired DNA replication, single strand DNA (ssDNA) exposure, DNA damage, nuclear structure disruption, and cell death[11–13], suggesting that protease activity of FAM111A needs to be tightly regulated for proper DNA replication and cellular homeostasis.

Faithful DNA replication is essential for maintaining genome integrity. However, replication fork stalling can occur at DNA damage,

and can lead to double-strand DNA breaks and genomic instability if it is prolonged[14–16]. To prevent fork collapse due to DNA damage, cells use mechanisms involving various DNA repair enzymes to facilitate DNA replication and repair at DNA damage sites. Our previous study implicated FAM111A in protection of replication forks at protein obstacles[10]. *FAM111A* KO cells exhibit replication fork stalling in the presence of camptothecin (CPT), a topoisomerase I (TOP1) inhibitor, or poly(ADP-ribose) polymerase inhibitors (PARPis)[10]. These inhibitors cause replication fork stalling by trapping the target enzymes on genomic DNA. Inhibition of TOP1 by CPT induces stabilization of topoisomerase 1 cleavage complexes (TOP1ccs), a reaction inter-mediate in which TOP1 is covalently bound to the 3′ end of the phos-phodiester backbone at a ssDNA break[17], while PARPis trap PARP enzymes at ssDNA break sites noncovalently[18,19]. Due to their bulky nature, DNA-protein crosslinks (DPCs) such as stabilized TOP1ccs or tight DNA-protein complexes including trapped PARPs block the movement of DNA polymerases, thereby imposing cytotoxicity on cancer cells[20]. Consistent with this notion, our previous study found that *FAM111A* KO cells are hypersensitive to CPT and PARPis, but not to cisplatin, which blocks DNA replication through interstrand DNA crosslinks[10]. Altogether, these findings suggest that FAM111A promotes DNA replication at protein obstacles.

FAM111A contains a serine protease domain (SPD, Fig. 1a) with a conserved catalytic triad (His385, Asp439, and Ser541). The FAM111A SPD belongs to the S1 family of serine peptidases, which includes chymotrypsin, in the MEROPS database[21,22]. The majority of the S1 family proteases are either extracellular or membrane-associated, but FAM111A is one of the few proteases in this family that are intracellular[23]. The protease activity of FAM111A is essential for its function of facilitating replication at stabilized TOP1ccs or trapped PARPs, as an active site mutant cannot rescue the replication defect phenotype of *FAM111A* KO cells when treated with CPT or PARPis[10]. However, while protease activity of FAM111A has been demonstrated in vitro using recombinant proteins through autocleavage activity[11], whether FAM111A is capable of cleaving proteins other than itself remains unknown.

Proteolysis of DPCs has emerged as an important mechanism for their repair. The metalloprotease SPRTN was the first protease to be identified to proteolyze DPCs at replication forks in higher eukaryotes[24–28]. To prevent nonspecific cleavage of cellular proteins, the activity of SPRTN is tightly regulated mainly through binding to ssDNA, which is present at stalled replication forks[24–27,29,30]. FAM111A has similar characteristics of protecting replication forks at protein obstacles, and it has been hypothesized that FAM111A proteolyzes DPCs[10]. However, its regulatory mechanism remains largely unknown, partly because the structure of FAM111A protein has not been deter-mined. Furthermore, there currently is no quantitative, in vitro assay to directly measure FAM111A protease activity, hampering the ability to examine the enzyme's activity.

Herein, we have developed an in vitro peptidase assay for FAM111A and characterized the SPD through structural studies to understand the regulatory mechanism of FAM111A. Our crystal struc-ture of the SPD has revealed that it is a dimer with a coiled coil interface between the N-terminal α-helices within the SPDs. We have engineered mutations to disrupt the dimer SPD interface to generate monomeric mutants. The crystal structure of the monomeric mutant revealed that the dimer is associated with a disorder-to-order transition that stabi-lizes the oxyanion hole at the active site. We have demonstrated that dimerization is necessary for protease activity in vitro as well as DNA replication at protein obstacles such as TOP1ccs stabilized by CPT in vivo. Furthermore, we establish that autocleavage can occur inde-pendent of FAM111A dimerization in our overexpression conditions. Collectively, these data suggest that dimerization is required for FAM111A's substrate-cleaving activity and its cellular functions but not for the autocleavage.

## Results

### SPD is sufficient for FAM111A peptidase activity

Recombinant proteins containing full-length human FAM111A were expressed in insect cells and purified with an N-terminal Strep tag (Fig. 1a, b). Additionally, recombinant SPD proteins, spanning residues from the previously reported autocleavage site[10] through the carboxy terminus (Fig. 1a), were generated by expression in *E. coli*, followed by affinity purification using the His.MBP tag, tag cleavage, and further purification via chromatography (Fig. 1b). We then evaluated peptidase activity of these proteins using a commercially available protease substrate consisting of a peptide with a C-terminal phenylalanine linked to a 7-amino-4-methyl coumarin (Suc-AAPF-AMC) (Fig. 1c). Peptidase activity was measured in real-time by monitoring the increase in AMC fluorescence after the substrate was mixed with enzymes. Wild-type (WT) FAM111A SPD exhibited a linear increase in AMC fluorescence over time in an SPD-concentration dependent manner while the active site mutant S541A produced no detectable change (Fig. 1d and Supplementary Fig. 1a), indicating that this sub-strate can be used to assay FAM111A peptidase activity. The wild-type full-length FAM111A protein also exhibited an increase in AMC fluor-escence over time, while the S541A mutant showed no increase (Fig. 1e). The specific activity of SPD was notably higher than that of full-length FAM111A (Fig. 1f), suggesting potential autoinhibition by the N-terminal region of FAM111A. Based on these results, we focused on the SPD-containing fragment for further analyses.

### FAM111A exhibits chymotrypsin-like protease activity

Serine proteases exhibit a preference for specific amino acids at the residue N-terminal to the cleavage site (designated as a P1 residue): chymotrypsin cleaves after Phe/Tyr/Trp residues; trypsin cleaves after Arg/Lys residues; and elastase cleaves after Ala/Val residues[31]. To determine the P1 specificity of FAM111A, we assayed the cleavage of three commercially available AMC fluorogenic peptide substrates. Wild-type SPD cleaved Suc-AAPF-AMC, which contains a phenylalanine at the P1 position, but did not cleave Boc-LRR-AMC, which has an arginine at P1, and MeOSuc-AAPV-AMC, which contains valine at P1 (Fig. 1g and Supplementary Table 1). Similarly, since serine proteases can be inhibited by substrate-mimic chloromethyl ketone (CMK) inhibitors[32–34], we used the sensitivity to CMK inhibitors to corroborate substrate specificity. FAM111A SPD was potently inhibited by N-*p*-Tosyl-L-Phe chloromethyl ketone (TPCK), which has phenylalanine in the P1 position (Fig. 1h). In contrast, the FAM111A SPD was not inhibited by MeOSuc-AAPV-chloromethyl ketone (SPCK), which has a valine at P1, and only weakly by Nᵅ-Tosyl-Lys chloromethyl ketone (TLCK), which has a lysine in the P1 position. Collectively, these data, along with the identification that FAM111A's site of autocleavage is also the C-terminal side of a phenylalanine[10], suggest that FAM111A prefers phenylalanine at the P1 site, similar to chymotrypsin.

Chymotrypsin and FAM111A SPD share a sequence identity of 16.8% (Supplementary Fig. 1b). To assess the similarity between the peptidase activity of FAM111A and chymotrypsin, we compared the activity of both enzymes using fluorescent peptide substrates con-taining Phe, Tyr, or Trp residues at the P1 position. FAM111A SPD exhibited peptidase activity with Phe (P1) substrates as mentioned above, and with Tyr (P1) and Trp (P1), although to a lesser extent (Supplementary Table 1). This trend mirrors chymotrypsin activity, although FAM111A exhibits a stronger preference for Phe at the P1 residue compared to chymotrypsin. These results suggest that FAM111A exhibits chymotrypsin-like activity in vitro, characterized by cleavage occurring C-terminal to a phenylalanine, a hallmark of the FAM111A protease.

### The serine protease domain of FAM111A forms a dimer

The SPD protein has a calculated molecular weight of 31.5 kDa. How-ever, the purified protein eluted from a size-exclusion

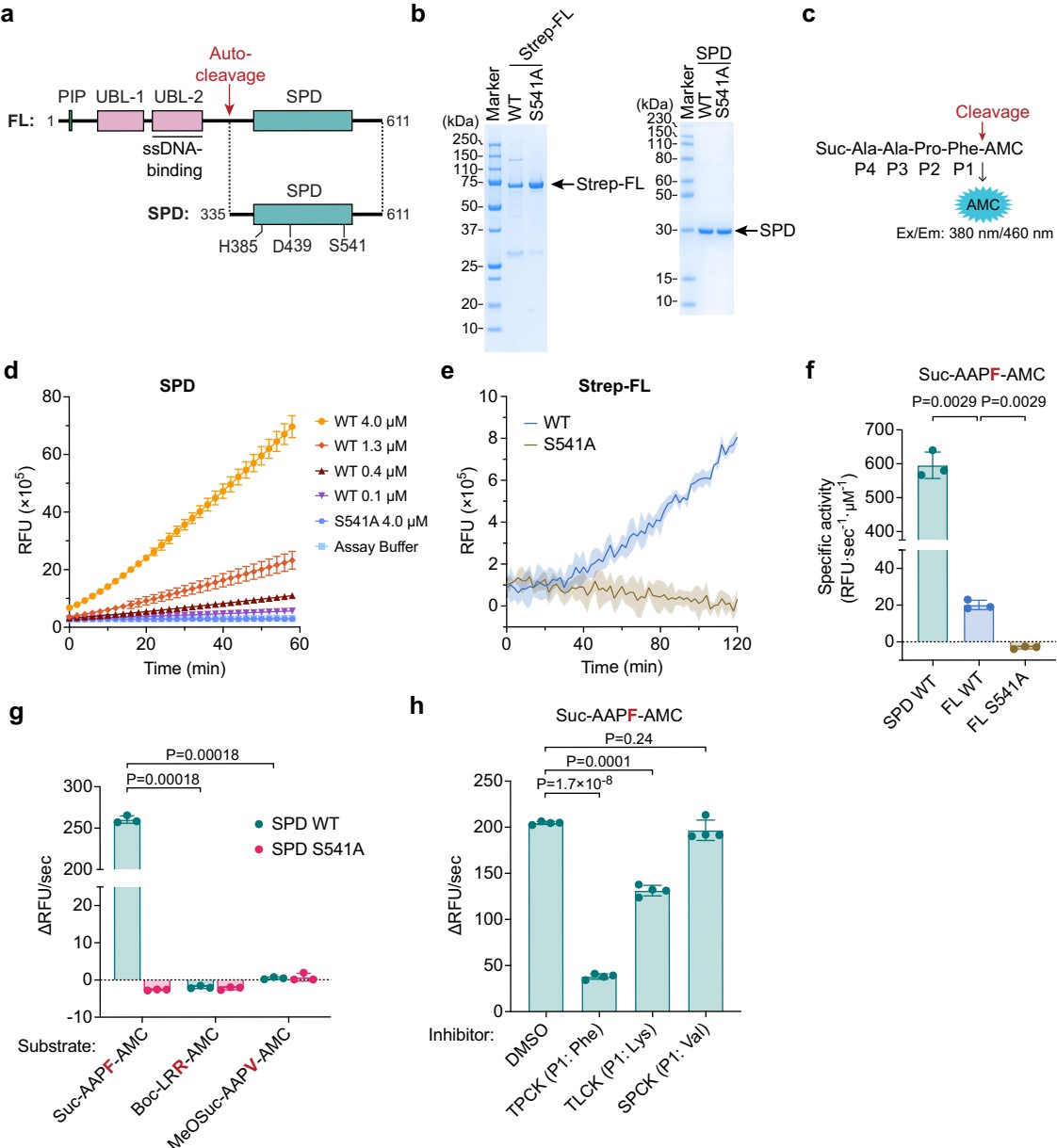

**Fig. 1 | FAM111A SPD exhibits chymotrypsin-like protease activity. a** Schematic representation of the FAM111A domain architecture. The autocleavage site and catalytic triad residues are indicated. FL: full length; PIP: PCNA-interacting peptide; UBL: ubiquitin-like; ssDNA: single-strand DNA; SPD: serine protease domain. **b** Purified recombinant Strep-FAM111A FL and SPD analyzed by SDS-PAGE. The sizes of molecular weight marker proteins are indicated on the left. Strep: Twin-Strep tag. **c** Schematic representation of the SPD protease assay. The AMC is attached to the substrate peptide at the C-terminus. Protease-catalyzed hydrolysis yields fluorescence from AMC, which is measured at Excitation/Emission at 380 nm/460 nm. **d** In vitro peptidase assay with WT SPD and S541A. Values are mean ± s.d. of three replicates. RFU: relative fluorescence units. **e** In vitro peptidase assay with WT Strep-FL and S541A. Values are mean of three replicates and shaded areas indicate s.d. **f** Specific activities of FAM111A FL and SPD. The enzyme reactions were performed with 0.5 μM SPD and 5.1 μM full-length FAM111A. Activity is represented as

the change in RFU over time. Values are mean ± s.d. of three replicates. **g** Peptidase activity of FAM111A SPD measured using various substrates. SPD activity was measured using chymotrypsin (Suc-AAPF-AMC), trypsin (Boc-LRR-AMC), and elastase (MeOSuc-AAPV-AMC) substrates. Values are mean ± s.d. of three replicates. **h** Inhibition of SPD by various serine protease inhibitors. Peptidase assays using FAM111A SPD were carried out as in (**d**) using Suc-AAPF-AMC as a substrate in the absence (DMSO) or presence of inhibitors (1 mM). Values are mean ± s.d. of four replicates. TPCK: Tosyl-L-phenylalanyl-chloromethane; TLCK: $N^{\alpha}$-Tosyl-Lys chloromethyl ketone, hydrochloride; SPCK: N-(methoxysuccinyl)-Ala-Ala-Pro-Val-chloromethyl ketone. In (**f–h**), significance of differences was determined by two-tailed unpaired t-test. Experiments in (**d, g**) were repeated three times, experiments in (**e**) more than three times, experiments in (**f, h**) twice with similar results. Source data are provided as a Source Data file.

chromatography (SEC) column at a volume that corresponds to an estimated molecular weight of 67 kDa (Fig. 2a and Supplementary Fig. 2a), suggesting that the 31.5 kDa SPD forms a stable dimer in solution. Dimerization was also validated by analytical ultracentrifugation (AUC) sedimentation velocity. The sedimentation coefficient for the SPD normalized for 20 °C in water was 3.835 S (20,

w) (Supplementary Fig. 2b), from which we determined a molar mass of 62,061 Da by using a calculated partial specific volume (v-bar) of 0.736 mL/g. This molecular mass closely matches that of an SPD dimer.

The dimerization of FAM111A SPD was unexpected, considering that the FAM111A SPD contains a trypsin-like fold, which is typically monomeric with the active site assembled from residues originally

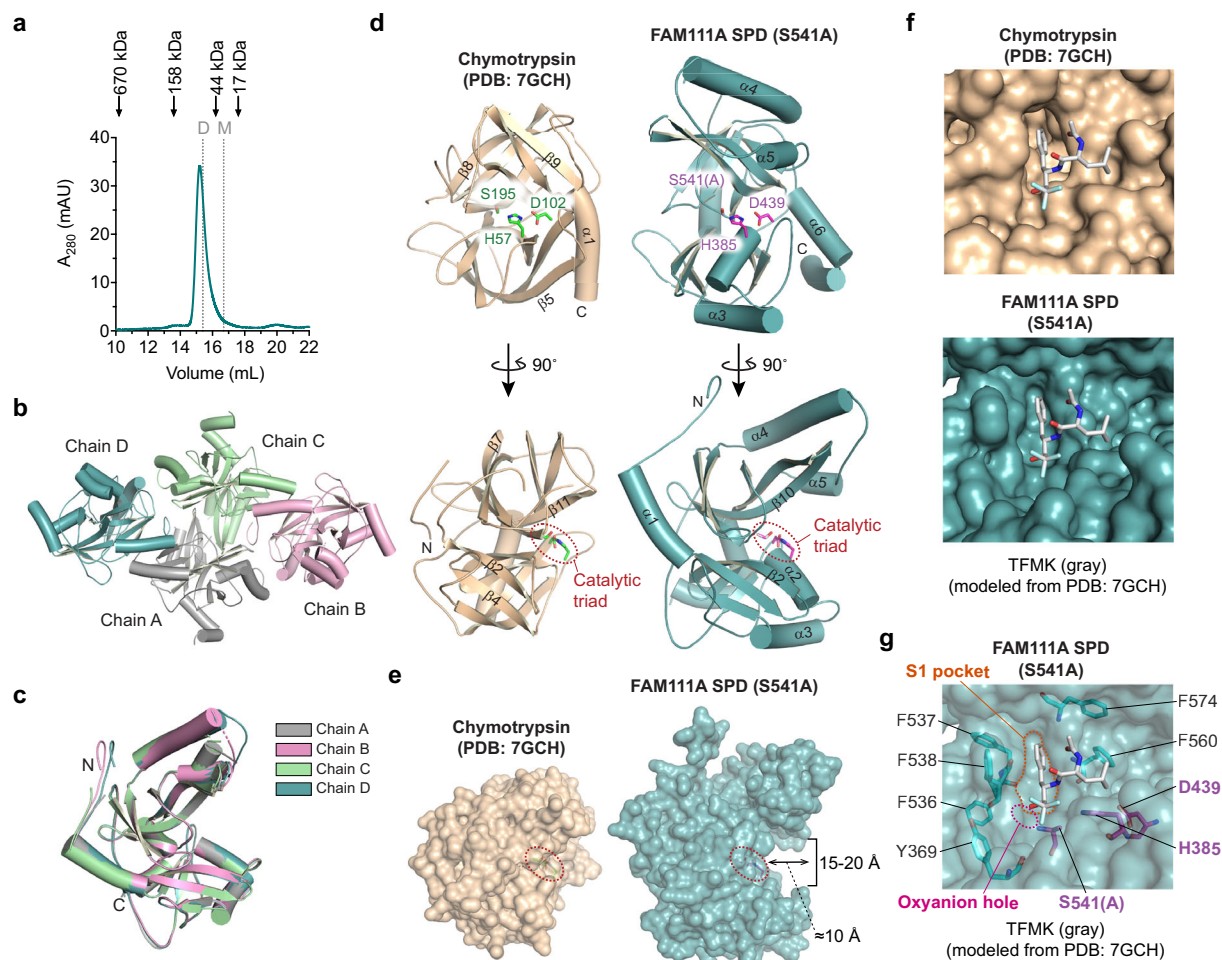

**Fig. 2 | Molecular architecture of FAM111A SPD. a** A representative chromatogram of size-exclusion chromatography for serine protease domain (SPD) S541A. Purified SPD S541A protein was loaded on Superdex 200 Increase 10/300 GL at 1 mg/mL. Absorbance at 280 nm (A280) is shown. Elution volumes for SEC standard proteins are indicated with arrows on top. Expected theoretical elution volumes for an SPD monomer (31.5 kDa, indicated with a letter M) and an SPD dimer (63.0 kDa, indicated with a letter D) are shown with dotted lines. **b**, **c** Crystal structure of SPD (S541A) containing four monomeric chains in the asymmetric unit (**b**) and an overlay of the four chains (**c**). **d** Two orthogonal views of chymotrypsin (PDB: 7GCH, left) and FAM111A SPD S541A (PDB: 8S9K, right). Side chains of the catalytic residues are shown and marked with red dotted lines in the bottom panels. **e** Space-filling models of chymotrypsin (PDB: 7GCH, left) and FAM111A SPD S541A (PDB: 8S9K, right). Side chains of the catalytic residues are shown and marked with red dotted lines. The dimensions of the trench in the FAM111A SPD are shown. **f** The trifluoromethyl ketone (TFMK) inhibitor (gray) modeled in the S1 site of FAM111A SPD (PDB: 8S9K, right) based on the structure of the TFMK-chymotrypsin complex (PDB: 7GCH, left). **g** The S1 pocket (outlined with dotted orange lines) of FAM111A SPD and surrounding hydrophobic residues are colored in cyan. The oxyanion hole is indicated with a dotted magenta line, and the catalytic triad residues are labeled in purple. The structure of TFMK (gray) is modeled based on PDB: 7GCH. Experiments in (**a**) were repeated more than three times and similar results were obtained.

derived from a single polypeptide chain. Therefore, we sought to determine the molecular basis of FAM111A dimerization and assess how dimerization could be related to its protease activity using X-ray crystallography. To mitigate the potential problem with protein production due to autocleavage, we generated the mutant SPD (S541A), which lacks the serine nucleophile of the catalytic triad, and performed crystallization trials with the purified protein. The structure was solved at 2.7 Å (Protein Data Bank (PDB) ID: 8S9K; Supplementary Table 2; Supplementary Figs. 2c and 3a) and contained four SPD proteins per asymmetric unit (Fig. 2b). Overall, the individual SPD chains are similar to each other (Fig. 2c), with root mean square derivations (RMSDs) ranging from 0.15 Å to 0.3 Å over Cα atoms. Thus, we focused our analysis on chain D, which is the most complete, but our analysis is broadly applicable to all chains.

The X-ray structure of the FAM111A SPD reveals an architecture similar to chymotrypsin (PDB ID: 7GCH)[35], featuring two β-barrels and a catalytic triad composed of Asp-His-Ser(Ala), with a calculated RMSD value of 2.2 Å (Fig. 2d). Interestingly, a protrusion of β-strands (β2 and β10) buttressed by helices α2 and α5 yields a protease domain with an active site located at the bottom of a trench, whereas the active site of chymotrypsin is shallow and accessible for substrate engagement (Fig. 2d, bottom and Fig. 2e). This suggests that the FAM111A SPD active site architecture is tailored to cleave smaller substrates, such as linker regions or disordered substrates, and not globular proteins.

An alignment of the FAM111A structure with that of chymotrypsin co-crystallized with a trifluoromethyl ketone (TFMK) transition state analog[35] (Fig. 2f, g) reveals that FAM111A shares key features with chymotrypsin. FAM111A possesses an oxyanion hole formed by the backbone amide nitrogen of Ser541 and Gly539, as well as a P1 residue binding pocket (designated as the S1 pocket) similar to that of chymotrypsin. This S1 pocket can accommodate phenylalanine, a preferred P1 residue (Fig. 1g, Supplementary Tables 1), as well as Phe334 present at the FAM111A autocleavage site[10]. While the S1 pocket in FAM111A is about 0.5 Å narrower than that observed in chymotrypsin, it can still easily accommodate phenylalanine by allowing rotation of the peptide bond between Phe537 and Phe538. The hydrophobic moieties that line the S1 pocket include Phe538, the main-chain backbones of 557-559, the α/β methylene of Glu573, and His556.

Furthermore, there are additional solvent-exposed hydrophobic residues in the vicinity of the S1 pocket (Tyr369, Phe536, Phe537, Phe538, Phe560, and Phe574) (Fig. 2g). This suggests that the presence of hydrophobic residues in substrates in the vicinity of the cleavage site may promote association with and cleavage by FAM111A.

## Identifying the dimer interface in FAM111A SPD

Since the active site of FAM111A SPD is contained within a single polypeptide, we sought to explain the curious observation that the SPD is a dimer, and how dimerization could be linked to the FAM111A function. We identified two potential dimer interfaces that are present between adjacent polypeptide chains in the crystal lattice. The first consists of a hydrophobic coiled coil interaction between the N-terminal alpha helix (α1) of SPDs (Fig. 3a), and the second was composed of main chain hydrogen bonds forming a β-sheet between β10 of adjacent chains (Fig. 3b).

To determine which interface corresponds to the dimer interface in solution, we engineered mutations that would specifically disrupt one dimer interface or the other and evaluated the oligomeric state of the mutant SPD proteins using SEC. To disrupt the interface at the α1 helix, we mutated hydrophobic residues Val347 and Val351 in the coiled coil interface to aspartic acid (D), (V347D and V351D, respectively) (Supplementary Fig. 3b), which would introduce a steric clash as well as charge repulsion across the α1 dimer (Fig. 3c). To disrupt the β10 interface, we generated a Thr563 to proline (P) (T563P) mutant, which would disrupt the β10 sheet hydrogen bonds (Fig. 3d and Supplementary Fig. 3b). We generated recombinant SPD containing these mutations (Supplementary Fig. 4a) and analyzed them at equal

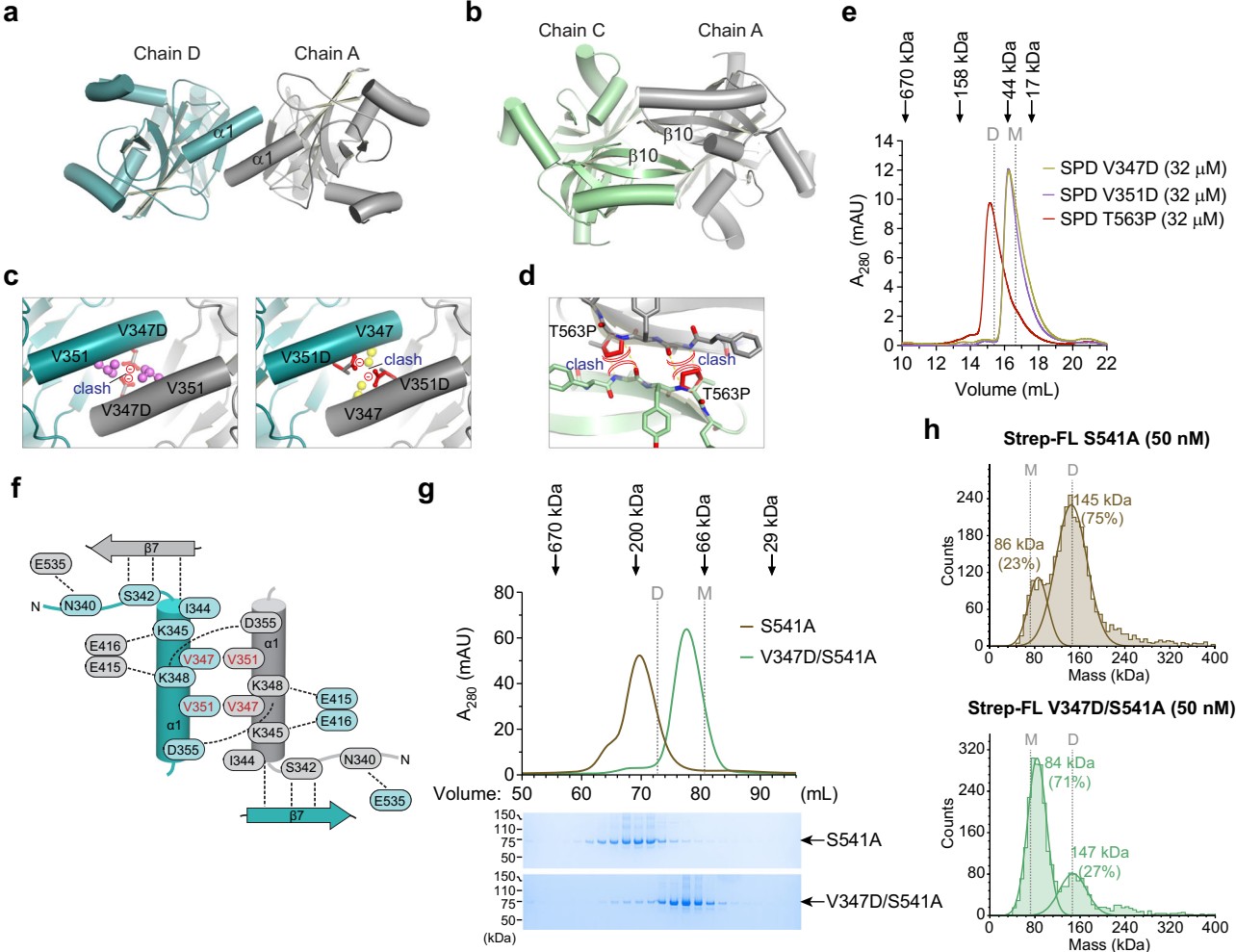

**Fig. 3 | Identification of the dimer interface in FAM111A SPD. a, b** Two possible dimer interfaces identified in the crystal lattice of FAM111A serine protease domain (SPD). Two subunits of a dimer, Chain A and Chain D in (**a**) and Chain A and Chain C in (**b**), are shown. **c, d** Zoomed-in-view of the interfaces between Chain A and Chain D (**c**) and between Chain A and Chain C (**d**). Key residues at the respective dimer interface are indicated in yellow (V347), and pink (V351). Clashes between mutated residues are shown with charge repulsion and steric hindrance indicated by minus signs and curved double lines, respectively. **e** Chromatograms of mutants (V347D, V351D, and T563P) from Superdex 200 Increase. Elution volumes for SEC standard proteins are indicated with arrows on top. Expected theoretical elution volumes for an SPD monomer (31.5 kDa, indicated with a letter M) and an SPD dimer (63.0 kDa, indicated with a letter D) are shown with dotted lines. **f** Schematic diagram of the SPD dimerization interface mediated by α1. Residues that comprise the dimer interface of one SPD monomer (teal) and the other (gray) related by a 2-fold non-crystallographic symmetry axis are depicted along with hydrogen bonds as black dashed lines. Residues mutated to disrupt dimer formation are labelled in red. **g** Chromatograms of FL Strep-FAM111A S541A and V347D/S541A from HiLoad Superdex 200. Elution volumes for SEC standard proteins are indicated with arrows on top. Expected theoretical elution volumes for a monomer (73.6 kDa, indicated with M) and a dimer (147.1 kDa, indicated with D) are shown with dotted lines. SDS-PAGE analyses of the corresponding fractions are shown with the positions of molecular markers. **h** Mass distribution for FL Strep-FAM111A S541A and V347D/S541A obtained by mass photometry. Expected theoretical molecular weights for a monomer (73.6 kDa, indicated with M) and a dimer (147.1 kDa, indicated with D) are shown with dotted lines. Percentages of counts and estimated molecular weights for the monomer and dimer peaks are indicated. Experiments in (**g, h**) were repeated twice with similar results. Source data are provided as a Source Data file.

amounts and concentrations (32 μM) using analytical SEC to determine their oligomeric state (Fig. 3e). The T563P mutant eluted at a size comparable to the WT SPD at an elution volume of 15.2 mL, whereas both the valine mutants (V347D and V351D) eluted from the column at a volume (16.4 mL) that best correlates with a monomer with no observable presence of dimer protein at this concentration. Therefore, this suggests that the interface between α1 mediates SPD dimerization in the solution. This interface contains a 1400 Å$^2$ of buried surface area consisting of an interaction about a 2-fold non-crystalline axis of symmetry that contains two sets of 7 hydrogen bonds (three of which form main-chain hydrogen bonds to β7) and three salt bridges that abut the coiled coil interaction between the α1 helices (Fig. 3f).

## Dimerization of full-length FAM111A requires the α1 helix of the SPD

We next used SEC to assess whether full-length FAM111A forms dimers and whether this dimerization depends on the interactions at the α1 helix within the SPD. The Strep-FAM111A S541A protein (Fig. 1b) eluted at a volume corresponding to an estimated molecular weight of 199 kDa, roughly matching the calculated molecular weight of the Strep-FAM111A dimer (Fig. 3g). In contrast, the Strep-FAM111A S541A/V347D mutant (Supplementary Fig. 4a) predominantly eluted at a volume corresponding to an estimated molecular weight of 96 kDa, close to the theoretical elution volume for the monomeric Strep-FAM111A protein (Fig. 3g). These findings strongly suggest that full-length FAM111A also exists in a dimeric form, and the V347D mutation disrupts this dimerization. This conclusion was further supported by mass photometry, a technique that measures the mass of individual biomolecules by analyzing their light-scattering properties[36]. The results indicated that the peak fractions from the FAM111A S541A and FAM111A V347D/S541A SEC predominantly contained dimer and monomer forms, respectively (Fig. 3h). Collectively, these results confirm that the α1 helix within the SPD serves as an important dimerization interface for full-length FAM111A.

To determine the dissociation constant ($K_d$) of FAM111A dimers, we prepared various dilutions of the Strep-FAM111A S541A protein and analyzed them by mass photometry (Supplementary Fig. 4b). Based on measurements at four different concentrations (10, 12.5, 25, and 50 nM), the average $K_d$ value for full-length FAM111A S541A was calculated to be around 4.8 nM. Additionally, specific activity of the SPD proteins was similar at various concentrations (49–4000 nM) at which peptidase activity was measurable (Supplementary Fig. 1a), suggesting that the $K_d$ value of the SPD dimer is ≤ 49 nM. Altogether, these results suggest that the dissociation constant of full-length FAM111A is estimated to be in the low nanomolar range.

## Structure of monomeric FAM111A SPD

To determine the molecular consequences of SPD dimer formation, we first sought to determine the structure of the monomeric form of the SPD to enable a comparison of the two states. We prepared recombinant SPD V351D (Supplementary Fig. 4a) and performed crystallization trials. We found that it still crystallized in the same space group as WT with the α1 interface, which is likely due to the extremely high concentrations of SPD used in crystallization trials (>300 μM). Of note, we observed evidence of residual weak dimer formation at a high concentration (188 μM) during SEC (Supplementary Fig. 5a), suggesting that the dimerization defect of this mutant could be overcome by very high protein concentrations. We were unable to produce sufficient concentrations of the V347D SPD for crystallization trials. Next, we generated a double V347D/V351D mutant SPD (Supplementary Figs. 3b and 4a), but although it eluted as a monomer in SEC (Supplementary Fig. 5a) and crystallized in a different space group, α1-mediated dimer interface was still present in the crystal lattice. Therefore, to generate a mutant SPD that would crystalize without forming the α1 dimer interface even at the very high

concentrations used for crystallization, we truncated the N-terminal SPD to remove two-thirds of the dimer interface and included an engineered V347D mutation to create a construct that we have named mini-SPD (Supplementary Figs. 3b and 4a). We found that mini-SPD remained monomeric even at the higher concentration of protein during SEC (Supplementary Fig. 5a). We then performed sparse matrix crystallization screens with the purified protein and obtained crystals that diffracted to 1.85 Å (PDB: 8S9L; Supplementary Table 2).

The mini-SPD crystallized with two polypeptide chains present in the asymmetric unit. Although helix α1 is still present and visible in the electron density map (Supplementary Fig. 5b), the dimerization interface observed in the full SPD dimer structure is noticeably absent. Overall, the mini-SPD and SPD dimer structures possess a significant degree of similarity with an overall RMSD value of 0.5 Å, but with some small yet impactful differences (Fig. 4a). The catalytic triad Asp-His-Ser (in this construct the catalytic Ser541 was retained as wildtype) is unchanged, which is expected as they are anchored on the rigid beta-barrel domain. In contrast, several disordered regions were noted in the mini-SPD where some of the residues comprising the dimerization interface, several surface-exposed loops, and the oxyanion hole are altered. Residues 413-422 that form a part of the loop L4 between β4 and β5 were disordered and not observed in the mini-SPD (Fig. 4b). Another region adjoining the β6-β7 loop (Loop L6, residues 474-480) is also disordered in one chain and altered compared to the dimeric SPD in the other chain (Fig. 4c). In turn, this region abuts residues 535-539, which include Phe538 that is part of the oxyanion hole (Fig. 4d). The 535–539 region surrounding the oxyanion hole is visible in the electron density map, but in a conformation that is different from the dimeric SPD with higher B-factors, and furthermore varies between the two polypeptide chains (Fig. 4d).

The differences observed between the two chains of the mini-SPD structure and the dimeric SPD as well as the higher B-factors observed for residues 536–538 encompassing the oxyanion hole and S1 pocket indicate that the monomeric SPD is more disordered. This suggests that dimerization of the SPD is associated with a disorder-to-order transition that supports the oxyanion hole residues in the conformation that is competent for catalysis. Biochemical support for this hypothesis comes from thermal shift data. Monomeric SPD generated by single- or double-point mutants, or the mini-SPD are all less stable than dimeric SPD at the concentrations assayed (Supplementary Fig. 5c), consistent with the dimeric state being more ordered and therefore more stable to thermal denaturation.

## SPD dimerization triggers disorder-to-order transition

The loop L4, which is disordered in the monomeric SPD (Fig. 4b), is structured in the dimeric SPD and forms salt bridges with Lys348 and Lys345 of the dimerization partner through Glu415 and Glu416 (Fig. 4e). Furthermore, the loop L4 contains Tyr414, which is positioned in the dimer structure adjacent to the loop L6 that is also unstructured in the monomeric SPD but structured in the dimeric SPD (Fig. 4e). This suggests that the loop L4 may function as a dimerization sensor that becomes ordered upon dimerization as well as a transducer of the structural changes initiated by dimerization. To evaluate the dynamics of the loop L4 and other regions of the SPD during dimerization, we performed molecular dynamics (MD) simulations to model the thermal motions of the protein that are missing from the static snapshots of the SPD yielded by the crystal structures. We performed an all-atom MD simulation over a course of 200 ns, starting from the crystal structures. The regions that are absent in the electron density in the mini-SPD are highly mobile whereas they adopt a stable structure in the dimeric form, consistent with the dimer formation associated with disorder-to-order transition. A plot of the root-mean-square fluctuation (RMSF) per residue shows distinct regions that are disordered in the monomeric SPD, including residues of the loops L4, L6,

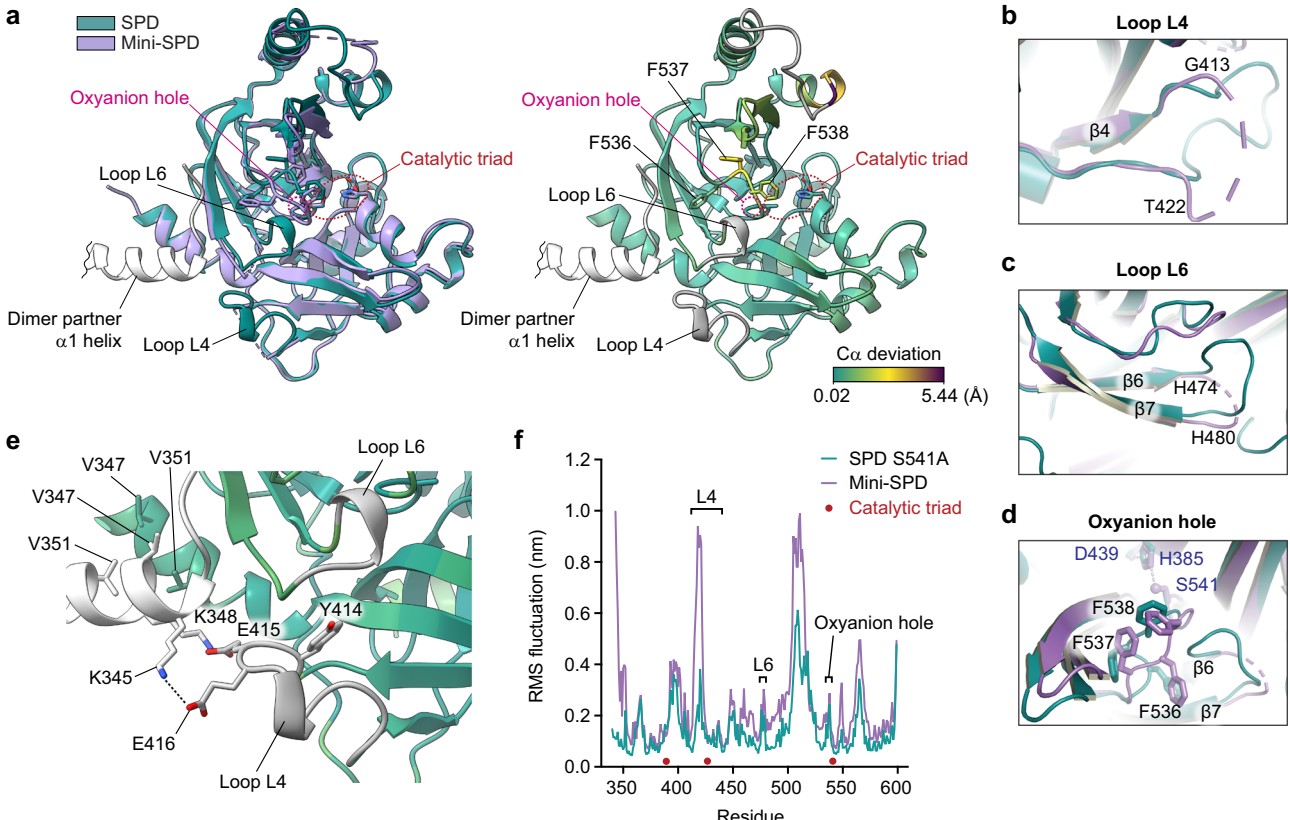

**Fig. 4 | Structure of a monomeric mutant of FAM111A SPD. a** A structural overlay of the serine protease domain (SPD) monomeric mutant (Mini-SPD) (PDB: 8S9L, lavender) with SPD S541A in the dimer arrangement (PDB: 8S9K, teal) (left). The structural model of SPD S541A was colored based on Cα deviation with mini-SPD, and the regions unstructured in mini-SPD are shown in gray (right). Side chains of the catalytic residues are shown and indicated by dotted red lines. The oxyanion hole is indicated with dotted magenta lines. Side chains of the three phenylalanine (F536, F537, and F538) that are disordered in mini-SPD are shown and labeled. The α1 helix of the dimer partner is shown in white. **b** The residues 413–422 within the loop L4 are structured in SPD (S541A) and disordered in mini-SPD (marked by a dashed line). SPD: teal; mini-SPD: lavender. **c** Loop L6 (residues 474–480) is structured in SPD (S541A), but it is disordered in mini-SPD (marked by a dashed line). SPD: teal; mini-SPD: lavender. **d** The disordered region in mini-SPD containing

Phe536, Phe537, and Phe538 is ordered in the dimer arrangement, stabilizing the oxyanion hole. Side chains of the catalytic residues (H385, D439, and S541) are shown and labeled in purple letters. **e** Interaction of the loop L4 with the α1 helix of the dimerization partner. The SPD S541A structure is colored as in (**a**). Glu415 and Glu416 on the loop L4 make a salt bridge with Lys348 and Lys345 on the α1 helix, respectively. Tyr414 in the loop L4 mutated in this study is shown. Side chains of Val347 and Val351 in the dimerization interface are also shown. **f** MD simulation analyses of WT SPD (teal) and the mini-SPD (lavender). The graphs show the calculated RMSF for each residue. The loops L4, L6, and oxyanion hole, which contain regions disordered in the monomeric mutant (mini-SPD) structure, are indicated with brackets. Positions of catalytic triad residues are indicated with red dots. Source data for the MD simulations are provided in Figshare [https://doi.org/10.6084/m9.figshare.24915123].

as well as the oxyanion hole being amongst the regions that change (Fig. 4f). These results suggest that the disorder-to-order transitions through the L4-L6 loops convey the effect of structural changes triggered by dimerization to the oxyanion hole.

## FAM111A SPD requires dimerization for protease activity

To determine whether dimer formation is critical for SPD activity, we next sought to define the role of dimerization in regulating the SPD catalytic activity by examining the activities of the WT SPD relative to the monomeric mutants in an in vitro peptidase assay using Suc-AAPF-AMC as a substrate (Fig. 5a). Although WT SPD exhibited robust catalytic activity, the activity of the monomeric mutant V347D was nearly undetectable. A small amount of residual activity was observed for V351D, which could be due to a small amount of residual dimerization, which can be detected at very high concentration (188 μM) by SEC (Supplementary Fig. 5a). The double mutant of V347D and V351D and mini-SPD, both of which exhibited diminished dimerization even at high concentrations, showed impaired and in the case of mini-SPD nearly undetectable activity (Fig. 5a). Furthermore, the activity of full-length Strep-FAM111A was also abolished by the V347D mutation

(Fig. 5b and Supplementary Fig. 4a), suggesting that dimerization is crucial for FAM111A activity.

To determine whether the residues involved in the dimer-sensing mechanism are also important for the activity of SPD, we mutated Lys348, which contacts Glu415 in the dimer sensor loop L4 of the dimer partner, and Tyr414, a residue on the dimerization sensor loop (Fig. 4e). Mutagenesis of Glu415 was unsuccessful due to poor protein stability of the mutant. SEC showed that dimerization is disrupted by the K348A mutation (Supplementary Fig. 5d), indicating that Lys348 contributes to the dimerization of SPD. The dimer-monomer equilibrium was also shifted slightly towards the monomer with the Y414A mutant (Supplementary Fig. 5d), although Tyr414 is not directly involved in the interaction with the dimer partner. Consistent with the requirement of SPD dimerization for the protease activity (Fig. 5a, b), peptidase assays showed reduced peptidase activity with the K348A mutant (Fig. 5c). Importantly, the Y414A mutation exhibited a much larger impact on peptidase activity than the more monomeric K348A mutant (Fig. 5c; Supplementary Fig. 5d). This might suggest that Tyr414 plays an additional role in the activation cascade. Collectively, these data demonstrate the importance of SPD dimerization for its activity and underscore the roles of the L4 loop as a dimerization

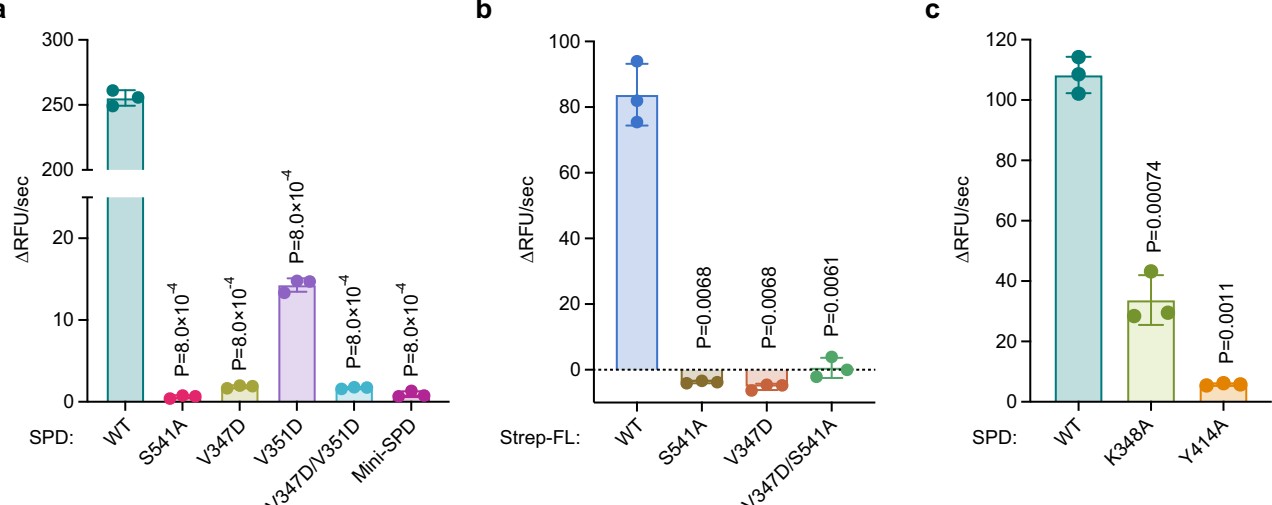

**Fig. 5 | Dimerization is required for the activity of FAM111A SPD. a** Protease assays of monomeric serine protease domain (SPD) mutants. Assays were performed with FAM111A SPD proteins (2 μM) using Suc-AAPF-AMC as a substrate. Activity is represented as the change in relative fluorescence units (RFU) over time. Values are mean ± s.d. of three replicates. **b** Protease assays of full-length FAM111A. Assays were performed with Strep-FL proteins (2.2 μM). **c** Protease assays of SPD harboring the K348A or Y414A mutations. In (**a**–**c**), values are mean ± s.d. of three replicates, and significance of differences between mutants and WT was determined by two-tailed unpaired t-test. Experiments in (**a**) were repeated three times and (**b**, **c**) twice with similar results. Source data are provided as a Source Data file.

sensor as well as a transducer of structural changes caused by dimerization.

## FAM111A dimerization is dispensable for autocleavage

Disease-associated mutations in FAM111A are known to result in increased autocleavage activity[10,11]. To assess how these mutations might alter FAM111A protease activity, we generated recombinant SPD proteins containing the KCS2 R569H or GCLEB D528G mutations (Supplementary Fig. 6a). Both mutants readily cleaved the Suc-AAPF-AMC substrate at a rate similar to that of the WT, with R569H hydrolyzing slightly faster and D528G slightly slower than WT (Fig. 6a). SEC of these patient-associated mutants suggests that they exist in a dimer form similar to WT SPD (Supplementary Fig. 6b). The peptidase activity of the R569H mutant was diminished by the V347D or the V351D mutation (Fig. 6b), suggesting that the patient-associated mutant still requires dimerization for the substrate cleavage activity.

We then explored whether the autocleavage activity of these patient-associated mutants also requires dimerization. We confirmed that the V347D mutation disrupts intermolecular interaction of full-length FAM111A in vivo by coimmunoprecipitation (Supplementary Fig. 6c). Surprisingly, the R569H patient mutant retained the autocleavage activity in vivo even when combined with the dimer-disrupting V347D mutation (Fig. 6c). Furthermore, the autocleavage activity of the R569H mutant was unaffected by the K348A and Y414A mutations (Fig. 6c). Similar results were observed for the hyper-autocleavage activity associated with the D528G mutant (Supplementary Fig. 6d), and with the basal level autocleavage activity of wild-type FAM111A (Supplementary Fig. 6e). Notably, the Y414A mutation not only had no effect on the autocleavage activity of the wild-type FAM111A but also stimulated it to a similar extent to the hyper-autocleavage activity of the R569H mutant (Supplementary Fig. 6e). Overall, these results suggest that autocleavage activity in both patient-associated mutants and wild-type FAM111A is governed by a distinct mechanism from substrate cleavage activity, as it does not strictly require dimerization or the key downstream residues in these overexpression conditions.

## Dimerization is important for FAM111A's function in DNA replication

Our previous study demonstrated that *FAM111A* KO causes the accumulation of TOP1ccs[10]. We therefore sought to test whether dimerization is important for the function of FAM111A in preventing TOP1cc accumulation in cells. To address this question, we expressed WT FAM111A, the dimer-interface mutants (V347D and V351D), or the dimerization sensing mechanism mutant (K348A), in the *FAM111A* KO cells (Fig. 7a, b). The Y414A mutant was not included in this experiment as its expression level was lower than WT, most likely due to its increased autocleavage activity (Supplementary Fig. 6e). The higher expression level of the V347D mutant, and possibly that of the V351D mutant, might reflect its severely impaired protease activity (Fig. 5b), which could reduce cytotoxicity. Expression of WT FAM111A prevented the spontaneous formation of TOP1cc foci in *FAM111A* KO, as described previously[10]. In contrast, the expression of the monomeric or the dimerization sensing mechanism mutant failed to prevent TOP1cc accumulation (Fig. 7c–f), despite their comparable or higher expression levels to that of WT (Fig. 7a, b) and their proper localization on chromatin (Supplementary Fig. 7a, b).

In the previous study, we have also reported that *FAM111A* KO cells exhibit replication fork stalling in the presence of CPT using DNA combing assays[10]. In this assay, nascent DNA is labeled with CIdU for 30 min and then with IdU in the presence or absence of CPT for 30 min (Fig. 7g). In the absence of CPT, there were no significant changes in the fork movement by the expression of WT, V347D, V351D, or K348A FAM111A in the KO cells. In the presence of CPT, on the other hand, ectopic expression of WT FAM111A rescued the replication fork defects as previously observed[10], while expression of the V347D, V351D, or K348A mutant did not (Fig. 7h, i). We attribute the partial rescue phenotype seen for V351D to the residual activity observed in vitro (Fig. 5a). This indicates that we observe a spectrum of fork-protection activity that correlates with defects in peptidase activity in vitro that are due to the disruption of dimerization. Altogether, these results suggest that mutations at the α1 helix of SPD disrupt FAM111A's ability to prevent TOP1cc accumulation as well as replication fork stalling in the presence of CPT, underscoring the importance of SPD dimerization in FAM111A's function in cells.

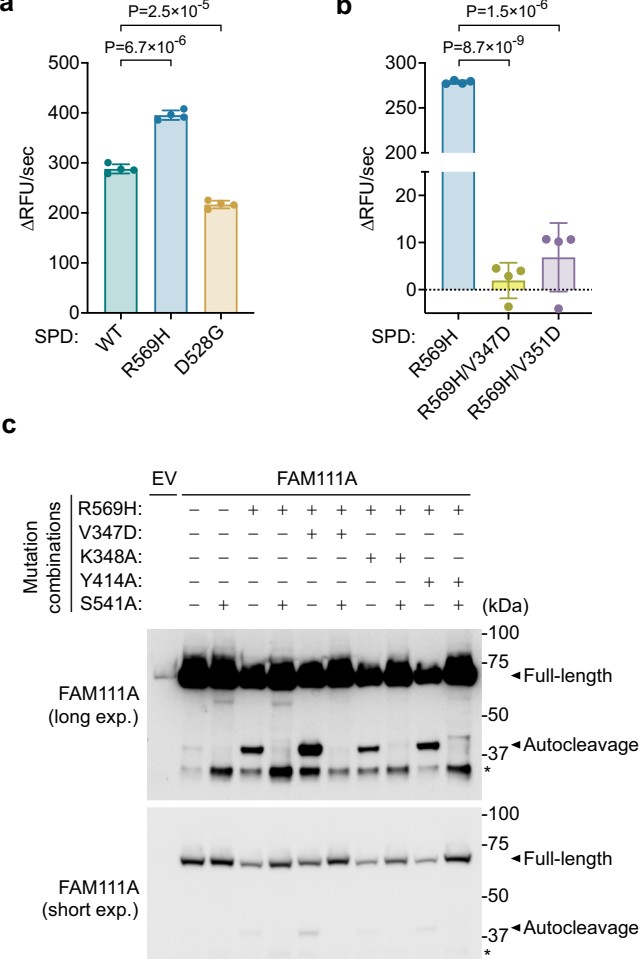

**Fig. 6 | FAM111A autocleavage is independent of dimerization. a** Peptidase activity of patient mutants serine protease domain (SPD) R569H and D528G measured using Suc-AAPF-AMC as a substrate. Assays were performed as in Fig. 1d with 4 µM SPD proteins. Activity is represented as the change in relative fluorescence units (RFU) over time. Values are mean ± s.d. of four replicates. Significance of differences between mutants and WT was determined by two-tailed unpaired t-test. **b** Peptidase activity of the indicated SPD mutants was measured as in (**a**). **c** FAM111A autocleavage. FAM111A proteins with indicated combinations of mutations were transiently expressed in 293 T cells (which lacks endogenous FAM111A expression) and examined by Western blotting. Long (top) and short (middle) exposures of anti-FAM111A blots are shown. β-actin (bottom) is shown as a loading control. The positions of full-length and autocleavage bands are indicated by black arrowheads. *Nonspecific degradation bands. Experiments in (**a**, **b**) were repeated twice and (**c**) three times with similar results. Source data are provided as a Source Data file.

## Discussion

FAM111A localizes at replication forks and promotes DNA replication at protein obstacles through its protease activity. However, little is known about the molecular architecture of this protease. In this study, using X-ray crystallography, biochemical analysis, MD simulations, and cellular assays, we demonstrate that the FAM111A SPD forms a dimer that is an active protease that supports replication in the presence of protein obstacles such as stabilized TOP1ccs. We have identified and described a series of SPD mutants that partially or fully disrupt dimerization and correspondingly yield enzymes that are partially or completely devoid of protease activity. We used one such mutant to determine the structure of the monomeric form of the SPD, which reveals an enzyme with an altered oxyanion hole. A comparison of the

monomeric and dimeric SPD structures and MD simulations suggests that dimerization of the SPD is linked to a disorder-to-order transition that plays a key role in the activation of the enzyme by stabilizing the conformation of oxyanion hole residues. Collectively, this study uncovers the structural basis of FAM111A protease activity, highlighting the importance of dimerization for proper enzymatic function and its cellular roles. These findings provide a better understanding of FAM111A's role in DNA replication and insights into the genetic disorders associated with *FAM111A* mutations.

In previous studies, FAM111A's protease activity was mainly investigated by measuring its autocleavage activity, partly due to the lack of proper enzyme assays in vitro. In our current study, we have developed an assay to measure FAM111A peptidase activity using a model peptide substrate. Through this assay, we show that FAM111A is capable of cleaving substrates other than itself and demonstrate the significance of dimerization for its activity. Our data indicate that the effects of dimer-disrupting mutations on in vitro peptidase activity correlate with their impact on FAM111A's ability to promote replication fork progression through TOP1ccs in cells. Altogether, these findings highlight the critical role of SPD dimerization in FAM111A's cellular function and confirm that the results of our in vitro peptidase assay accurately reflect the protease activity essential for FAM111A's in vivo functions.

Since FAM111A is a protease, it needs to be tightly regulated to prevent non-specific cleavage of itself and other essential cellular proteins. SPRTN has been shown to dimerize[29] and interestingly, dimerization regulatory mechanisms have also been reported for some viral serine proteases[37,38]. This suggests that dimerization may be a widely used mechanism for regulating protease activity because viruses also need to restrict protease activity to the correct phase of their life cycle. For any protease present in the sea of cellular proteins, mechanisms to prevent off-target proteolysis while still permitting degradation of their target will be critical. Proteases that are inactive as monomers can thus be safely present in cells with minimal risk of off-target proteolytic activity. Another possible mechanism for restricting FAM111A's activity could involve autoinhibition by its N-terminal domains. Supporting this notion, our enzyme assays demonstrated that the FAM111A fragment lacking the N-terminal region exhibits significantly higher specific activity compared to full-length FAM111A. In the future, it will be important to determine how additional domains, including the PIP box, UBLs, and ssDNA-binding domain (Fig. 1a), as well as possible FAM111A-interacting factors in cells influence proteolysis by FAM111A.

Both FAM111A and SPRTN share a topology where the active site is located at the bottom of a trench. Recent findings showed that SPRTN can specifically cleave substrates with a long flexible region rather than globular proteins[29] due to a narrow active site. A similarly recessed active site in FAM111A also suggests a similar substrate requirement. Of note, we expressed and purified the FAM111A SPD and did not observe the recombinant SPD protein cleaving itself to a noticeable extent, nor did we observe a time-dependent decrease in peptidase activity in our in vitro assays that would indicate autocleavage within the enzyme domain and inactivation. In contrast, the active site of chymotrypsin is much more accessible, making it well suited for a role in digestion where it would be advantageous to be able to proteolyze globular proteins. This suggests that a recessed active site that restricts access to globular proteins may be an important feature for DNA replication and repair proteases. However, while this feature may safeguard against non-specific degradation of other proteins, it poses a challenge if FAM111A's target proteins are globular. In the case of SPRTN, unfolding of tightly-folded proteins by the ATP-dependent unfoldase p97/VCP can enhance their degradation by SPRTN[39]. Whether a similar mechanism exists for FAM111A is an important question for future study.

This study also sheds light on the effects of FAM111A mutations linked to KCS2 and GCLEB. While it was initially assumed

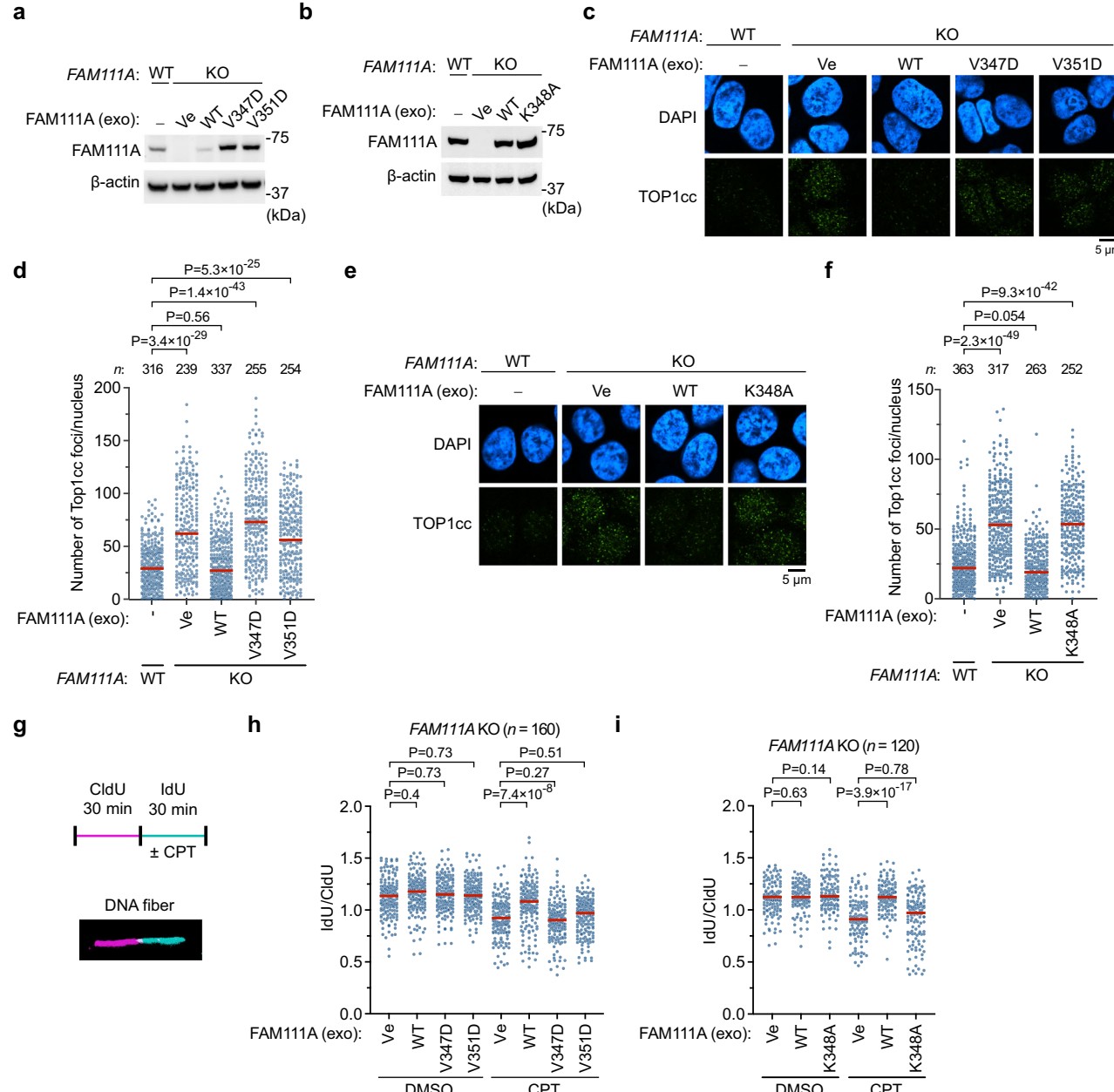

**Fig. 7 | Dimerization of FAM111A SPD is important for preventing TOP1cc accumulation and replication fork stalling. a, b** Exogenous (exo) expression of FAM111A in *FAM111A* knockout (KO) cells. The indicated FAM111A proteins were stably expressed by lentiviral vectors in *FAM111A* KO HAP1 cells and analyzed by Western blotting. β-actin is shown as a loading control. Ve: empty vector. **c–f** Topoisomerase 1 cleavage complex (TOP1cc) focus formation. Representative single z-slice images of cells stained with anti-TOP1cc antibody (green) and DAPI (blue) are shown in (**c, e**). Dot plots of TOP1cc foci quantification are shown in (**d, f**). Distribution of the number of TOP1cc foci per nucleus is plotted (red line: median). Significance of differences between experimental and control (*FAM111A* WT) samples was determined by two-tailed unpaired t-test. *n* is the number of individual measures. **g** A graphical representation of the DNA combing assay used to measure replication fork progression. Cells were labeled with CldU (red) for 30 min followed

by IdU (green) for 30 min. During the IdU labeling, dimethyl sulfoxide (DMSO) or camptothecin (CPT, 30 nM) was added (upper panel). The CldU and IdU tracks are visualized by immunostaining and the track lengths were measured. An illustrative picture of a replication track is shown (lower panel). **h, i** A dot plot showing distribution of IdU/CldU track length ratios in the presence and absence of CPT (red line: median). Replication tracts of *FAM111A* KO cells expressing WT, the monomeric mutants (V347D and V351D) in (**h**), and the K348A mutant in (**i**) were measured by an investigator blinded to sample identity. Significance of differences between experimental and control (Ve) samples was determined by two-tailed unpaired t-test. *n* is the number of individual measures. Experiments in (**c–f**) and (**h, i**) were repeated twice and similar results were obtained. Source data are provided as a Source Data file.

that patient-associated mutations lead to enhanced protease activity in mutant FAM111A, our data challenges this view. Specifically, our findings reveal that mutant SPD proteins with patient-associated mutations do not consistently exhibit a proportional increase in peptidase activity compared to the observed hyper-autocleavage activity in cells. Furthermore, we observed that, unlike its substrate-

cleaving activity, FAM111A's autocleavage activity does not strictly depend on dimerization in our assay condition using FAM111A overexpression. This suggests that the mechanisms governing substrate cleavage and autocleavage by FAM111A might be distinct. We note that the link between dimerization and autocleavage may be more complex than that dictated by Le Chatalier's principle, as the

net effect in a biological system may be due to the complex contributions of many other factors. Considering these findings, our data suggest that the primary impact of these mutations is hyper-autocleavage, rather than hyperactivation of FAM111A's normal protease function. Notably, hyper-autocleavage could still result in increased FAM111A activation, considering the possible auto-inhibitory function of the N-terminal region of FAM111A (Fig. 1f). By cleaving between the autoinhibitory domain and the SPD, hyper-autocleavage of FAM111A may indirectly result in unregulated FAM111A activity. Alternatively, these mutations could make the SPD resistant to the autoinhibition mediated by the FAM111A N-terminal region. Therefore, understanding the molecular consequences of these disease-associated FAM111A mutations is an essential area for future investigation.

FAM111A can protect cancer cells from chemotherapeutic drugs such as TOP1 inhibitors, which generate DPCs containing stabilized TOP1ccs[10]. Therefore, an inhibitor of FAM111A could be used to mitigate tumor drug resistance. FAM111A inhibitors could also be used to sensitize cells to DPC-inducing drugs or to enhance their efficacy in cancer cells where alternative pathways of DPC bypass and repair are missing or inhibited. However, the development of an active site inhibitor may be challenging, as the active site of FAM111A SPD shares core features with chymotrypsin, and likely other important cellular proteases. Thus, the general strategy of developing an active site inhibitor could lead to side effects and undesired toxicity due to off-target binding. Notably, the structure of monomeric FAM111A SPD reported here reveals surface-accessible regions that could be binding sites for an inhibitor that blocks dimerization or prevents the disorder-to-order transition associated with dimerization. Since these binding sites would not be conserved on other proteases, it would provide an opportunity for developing an inhibitor that is specific to FAM111A with the potential to enhance cancer therapeutic efficacy as well as treat disease caused by FAM111A misregulation.

# Methods

## Chemicals and reagents

All chemicals and reagents were purchased through Cayman, Sigma, and Thermo Fisher unless otherwise stated. MeOSuc-Ala-Ala-Pro-Val-AMC · Cayman (#14907); Boc-Lys-Arg-Arg-AMC · Cayman (#26642); Suc-Ala-Ala-Pro-Phe-AMC · Sigma (230914); Suc-Leu-Tyr-AMC · Cayman (#10008120); Ac-Ala-Asn-Trp-AMC · Fisher Scientific (#50-196-5076). Inhibitors were purchased through Sigma: TPCK (Tosyl-L-phenylalanyl-chloromethane) (#T4376); SPCK (N-(methoxysuccinyl)-Ala-Ala-Pro-Val-chloromethyl ketone) (#M0398); TLCK (Nα-Tosyl-Lys chloromethyl ketone, hydrochloride) (#616382). TCEP (Tris (2-carboxyethyl) phosphine) · Goldbio (#51805-45-9).

## Expression constructs

The cDNA containing human FAM111A was amplified by reverse transcription PCR and cloned into pLVX2-IRES-puro (no epitope tag) for lentiviral or transient expression in human cells. For expression of FAM111A with a C-terminal Flag or HA tag, the *FAM111A* open reading frame sequence was amplified using PCR primers containing an in-frame tag sequence and cloned in pLVX2-IRES-puro. For bacterial expression, a DNA fragment encoding FAM111A 335-611 was codon-optimized for *E. coli* expression and inserted into the pDB.His.MBP vector (with an N-terminal His6-MBP tag) for recombinant protein expression. For insect cell expression, full-length FAM111A cDNA codon-optimized for insect cells was inserted in pFastBac containing an N-terminal Twin-Strep tag. Recombinant bacmids were generated using the Bac-to-Bac Baculovirus Expression System following manufacturer's instructions (Thermo Fisher). Point mutations were introduced by Gibson assembly (NEB, #E2611S) or QuickChange (Agilent) and confirmed by Sanger sequencing.

## Mammalian cell culture

Human chronic myelogenous leukemia cell line HAP1 was obtained from Horizon Discovery (#C631) and cultured in Iscove's Modified Dulbecco's Medium supplemented with 10% fetal bovine serum (FBS). The *FAM111A* knockout HAP1 cell line (Clone #14) was generated previously using CRISPR/Cas9 and contains a 45-bp insertion with a stop codon in the *FAM111A* gene exon 4[10]. Human embryonic kidney cell line 293 T was purchased from American Type Culture Collection (#CRL-11268) and cultured in Dulbecco's modified Eagle's medium supplemented with 10% FBS. For transient protein expression in 293 T, cells were transfected with plasmids using Lipofectamine 2000 and harvested after 2 days.

## Recombinant protein expression

For bacterial expression, BL21(DE3) and Rosetta *E. coli* cells transformed with pDB.His.MBP-FAM111A SPD were cultured in Terrific Broth using a Lex-48 Bioreactor (Ephiphyte3) at 37 °C until an optimal $OD_{600}$ of 3-5 was achieved. Protein expression was then induced by the addition of 4% (v/v) glycerol, 2% (v/v) ethanol and 150 µM of IPTG with further incubation at 10 °C for 40 h. Cells were harvested using a Lynx 4000 centrifuge (Thermo Fisher) at 6000 x g for 20 min, and cell pellets were transferred to 50 mL Falcon tubes and stored at −80 °C until ready for protein extraction. For insect cell expression, bacmids were transfected into ExpiSf9 using ExpiFectamine Sf (Thermo Fisher) and virus-containing supernatants were collected after 4−5 days. ExpiSf9 cells cultured in ExpiSf CD medium supplemented with ExpiSf Enhancer (Thermo Fisher) were infected with virus stocks as instructed by the manufacturer and cultured for 2−3 days at 27.5 °C. Collected cells were washed once with PBS, frozen in liquid $N_2$, and stored at −80 °C until protein purification.

## Purification of recombinant protein

*E. coli* pellets frozen at −80 °C were thawed and resuspended in lysis buffer (50 mM sodium phosphate, pH7.5, 500 mM NaCl, 5 mM imidazole, 5% glycerol, 0.5 mM TCEP), then transferred to a chilled metal beaker. Cell lysates were sonicated at 80% power on a Branson 250 sonicator in three 30 s intervals, with 1-min cooling in between. Clarified lysate was collected after centrifugation at 25,000 x g for 30 min and passed over a Ni-NTA resin bed that was equilibrated in a lysis buffer stated previously. After washing three times with wash buffer (50 mM sodium phosphate, pH7.5, 500 mM NaCl, 20 mM imidazole, 5% glycerol, and 0.5 mM TCEP) to remove remaining cell debris or unbound proteins, proteins were eluted using an elution buffer (50 mM sodium phosphate, pH7.5, 500 mM NaCl, 250 mM imidazole, 5% glycerol, 0.5 mM TCEP). Elution from Ni-NTA was passed over amylose resin equilibrated with MS300 buffer (20 mM Tris pH 7.5, 300 mM NaCl, 0.5 mM TCEP). After washing four times with MS300 buffer, the His6-MBP-SPD protein was eluted using MS300 buffer containing 10 mM maltose. Eluted protein was subjected to precipitation with 2 volumes of 4 M ammonium sulfate at 4 °C overnight followed by centrifugation at 25,000 x g for 30 minutes at 4 °C. The precipitated protein was purified using FPLC immediately or kept at −80 °C until further use.

The precipitated protein pellets were re-dissolved by the addition of a minimal amount of MS300, then centrifuged at 20,000 x g for 10 min at 4 °C to eliminate any insoluble particulates. The clarified solution was injected into an ÄKTA go FPLC system (Cytiva) and purified on a HiLoad 16/600 Superdex 200 pg column (Cytiva) using MS300 buffer at a flow rate of 1 mL/min. His6.MBP-FAM111A were eluted, cleaved with a homemade TEV protease (150 µM) overnight, and run on a 6 mL RESOURCE Q column (Cytiva) with high salt (20 mM Tris pH 7.5, 1 M NaCl) and low salt (20 mM Tris pH 7.5, 3 mM DTT) buffers. To determine the oligomeric state of SPD, 100 µL of 1 mg/mL proteins were analyzed on a Superdex 200 Increase 10/300 GL column in MS300 buffer using ÄKTA pure FPLC system with

Unicorn (version 7.3) (Cytiva). The molecular weight of the elusted protein was estimated using a set of molecular mass standards (Bio-Rad, 1511901). Elution fractions from each column were tested on a Coomassie blue-stained SDS-PAGE gel. Fractions containing FAM111A protein were pooled together, and the protein was concentrated using 10 K cut-off centrifugal filters (Millipore). The purified proteins were stored in a 25% glycerol storage buffer (20 mM Tris pH 7.5, 300 mM NaCl, 1 mM TCEP) at −80 °C for use in downstream assays.

For purification of full-length FAM111A proteins produced in insect cells, cells were lysed in lysis buffer (50 mM HEPES-NaOH pH 7.5, 1 M NaCl, 1 mM $MgCl_2$, 10% glycerol, 1% NP-40) supplemented with 4 U/mL Benzonase (Novagen), 1 mM DTT, and 2 μM Pepstatin. Lysates were incubated on ice for 30 min, sonicated and cleared by centrifugation (20,000 × g, 4 °C, 30 min). Lysates were mixed with Strep-Tactin superflow resin (Qiagen) equilibrated with lysis buffer by rotation at 4 °C for 3 h. Resin was washed twice with lysis buffer and once with wash buffer (50 mM HEPES-NaOH pH 7.5, 250 mM NaCl, 10% glycerol). Purified proteins were eluted with elution buffer (wash buffer supplemented with 10 mM d-Desthiobiotin) and used for protease assays or stored at −80 °C in small aliquots. To determine the oligomeric state of full-length FAM111A, 500 μL of 1.6 mg/mL proteins were analyzed on a HiLoad 16/600 Superdex 200 pg SEC column in MS300 buffer supplemented with 0.2% 3-[(3-Cholamidopropyl)dimethylammonio]−1-propanesulfonate using the ÄKTApurifier FPLC system with Unicorn (version 5.11) (Cytiva). Molecular weights of the eluted proteins were estimated using the following proteins as standards: thyroglobulin (Sigma, #T9145), bovine serum albumin (Sigma, #A8531), β-Amylase (Sigma, #A8781), carbonic Anhydrase (Sigma, #C7025).

## Protein crystallization and structure determination

FPLC-purified FAM111A SPD and mini-SPD were concentrated to 7 mg/mL and 18 mg/mL, respectively, using a 10 K cutoff spin concentrator (Millipore). Crystals of FAM111A were grown using the sitting-drop vapor diffusion method by mixing 100 nL of protein and 100 nL of precipitant. The reservoir contained 25 μL of each crystallization solution. The JCSG++ and PACT (Jena Bioscience) sparse matrix screens were used to screen for crystallization conditions. Crystals of SPD were obtained at 20 °C in 20% (w/v) PEG 8,000, 100 mM Tris pH 8.5, 200 mM $MgCl_2$ grown at room temperature. Mini-SPD crystals were obtained at 4 °C in 100 mM BIS-TRIS, pH 5.5, and 2 M Ammonium sulfate. These crystals were transferred into a cryoprotectant solution containing 22% (w/v) PEG 8,000, 200 mM $MgCl_2$ pH 8, 16% (w/v) glycerol, and 100 mM BIS-TRIS pH 5.5, 2.2 M Ammonium sulfate, 25% glycerol respectively. All crystals were flash-frozen in liquid nitrogen prior to data collection. X-ray diffraction datasets were collected at the Advanced Photon Source at the NE-CAT beamlines (24-C and 24-E) using beamline in house software. X-ray diffraction datasets were processed and scaled using HKL2000 (version 720)[40]. The SPD S541A structure was solved using molecular replacement with the AlphaFold[41,42] predicted model of FAM111A residues 340-599 and the PHENIX (version 1.17.1-3660)[43], which yielded solutions with a TFZ score of 33.4 and LLG of 1309, indicating a strong solution. Similarly, the SPD monomer was used as a search model to solve the mini-SPD structure, which yielded a solution with a TFZ score of 36.3 and LLG of 1285. Interactive rounds of model building in Coot (version 0.8.9.1)[44] and refinement in PHENIX.REFINE against the high-resolution datasets were used to produce the final models. Diagrams of protein structure were generated using PyMOL (version 2.5.2, Schrodinger) or ChimeraX (version 1.6.1)[45]. Modelling, design of point mutations, and RMSD calculations were performed using PyMOL.

## Analytical ultracentrifugation sedimentation velocity

Analytical ultracentrifugation sedimentation velocity was carried out at 20 °C on an Optima analytical ultracentrifuge (Beckman-Coulter) using an 8-hole AN-50 Ti rotor with absorbance detection at 280 nm. Protein samples (0.9 O.D.) in 20 mM Tris pH 7.5, 300 mM NaCl, 0.5 mM TCEP were loaded into AUC cells composed of a 12 mm epon charcoal 2-sector centerpiece with quartz windows. The cells were first subjected to 3 h at 20 °C under vacuum to allow temperature equilibration, then the experiment was performed at 45,000 rpm with continuous data acquisition for 7 h (200 scans). The protein partial specific volume (0.736 mL/g) and buffer density and viscosity (1.0052 g/mL and 0.01002 poise respectively) were determined using SEDNTERP[46]. The data were analyzed using a continuous c(s) distribution in the program Sedfit (version 15.01b)[47] with fitting of frictional ratio (1.437), meniscus, time invariant noise and radial invariant noise.

## Mass photometry

Mass Photometry was performed using Refeyn Two MP system (Refeyn) following previously described methods[48]. Data acquisition and analysis were carried out using Acquire MP and Discover MP (2023 R1), respectively. Dissociation constants ($K_d$) of FAM111A dimers were calculated as described previously[49]. In brief, protein stocks were diluted with Phosphate Buffered Saline (pH 7.4) to final concentrations of 10, 12.5, 25, or 50 nM in low-protein binding tubes one hour prior to measurements to allow for monomer-dimer equilibration. After identifying the focal position using Acquire MP (2023 R1), 10 μL of a pre-diluted protein sample was directly applied onto a glass slide, and sixty-second movies were recorded for each measurement. Three measurements were taken for each concentration, and $K_d$ values were calculated based on the counts corresponding to the monomer and dimer species.

## Protease assays

Protease assays for SPD were performed in 96-well plates with a clear, flat bottom (Corning 3595). Protein samples were diluted to starting concentrations in a 25% glycerol storage buffer (20 mM Tris pH 7.5, 300 mM NaCl, and 1 mM TCEP). Protease reactions (50 μL) contained a substrate buffer (1 mM of the indicated substrate, 0.2 mg/mL BSA in the MS300 buffer) and FAM111A protein or bovine chymotrypsin (Promega). Unless otherwise indicated, 4 μM SPD proteins were used for peptidase assays. Protease was added last to initiate the reaction, and the plate was placed in a CLARIOstarPlus plate reader (BMG Labtech) that was pre-warmed to 37 °C. Measurements were recorded using Clariostar (version 5.61) with Ex/Em: 380 nm/460 nm every two minutes for 30 cycles. Control reactions containing only 25% glycerol storage buffer in place of FAM111A protein samples were used to correct for measurable background. Protease activity was calculated in MARS (version 3.41, BMG Labtech) by taking the slope of the change in fluorescence (ΔRFU/sec) over all cycles of each experiment. Protease assays on full-length FAM111A were performed essentially as described for SPD, except in a 25 μL reaction volume in a 384-well plate, and measurements were taken every two minutes for 60 cycles in the SpectraMax i3x plate reader (Molecular Devices) using SoftMax Pro (version 7.0.3). Unless otherwise stated, 5.1 μM full-length FAM111A proteins were used for peptidase assays.

## Thermal shift assay

Assays were performed with the Protein Thermal Shift™ Dye Kit (Applied Biosystems, cat. 4461146) according to the manufacturer's instructions. Briefly, the protein was diluted to 1 mg/mL and 12.5 μL of protein were added to 7.5 μL of assay solution (2.5 μL 1X SYPRO Orange dye, 5 μL thermal shift buffer) on ice for a total reaction volume of 20 μL. 25% glycerol storage buffer (20 mM Tris pH 7.5, 300 mM NaCl, 1 mM TCEP) was used as a non-protein control. Reactions were run on a CFX384 Touch Real-Time PCR System (Bio-Rad), and the data were collected using CFX Manager (version 3.1). The temperature was increased in a step-and-hold manner from 25 °C to 99 °C in a 0.05 °C/cycle increment and with an equilibration time of 2 min at each temperature. The HEX channel was used to analyze the resulting data to

correlate with the emission-excitation spectra of the SYPRO Orange dye (490 nm excitation, 624 nm emission). The melting temperature was defined as the first derivative of the peak of the melting curve.

## Molecular dynamics simulations

The crystal structures of FAM111A monomer (mini-SPD) and dimer were processed before performing molecular dynamics (MD) simulations, including stripping the water molecules, modeling missing residues in loops using Rosetta (version 2019.3-11)[50], and renumbering residues with Coot (version 0.8.9.1)[51]. GROMACS (version 2021.4) patched with PLUMED[51] was used for the MD simulations and data analysis. The processed FAM111A molecules were solvated in a periodic cubic box of length 9.854 nm for the dimer and 8.658 nm for the monomer with the four-point TIP4P rigid water model[52] under the force field OPLS-AA/M (version 2015)[53]. Protein charges were neutralized by adding sodium ions as counterions and sodium chloride (150 mM) was introduced to mimic physiological conditions. Energy minimization was used to remove steric clashes under the steepest decent algorithm until the maximum force was less than 200 kJ/mol/nm. A 200 ps constant volume/temperature (NVT) simulation and a 200 ps constant pressure/temperature (NPT) simulation with restrained heavy atom positions were used to equilibrate the systems to 300 K with a V-rescale thermostat[54] and to 1.0 bar with a Berendsen barostat[55]. The LINCS algorithm[56] was used to constrain all bond lengths with a time step of 2 fs. Replica Exchange with Solute Tempering (REST2)[57] simulation with leap-frog integration[58] was performed to simulate the aqueous behavior of the FAM111A monomer and dimer with the "hot atoms" setting for all the protein atoms, but without any position restraints. Twenty replicas with temperature spanning from 300 K to 600 K were performed with a Hamiltonian replica exchange attempt between adjacent replicas every 500 steps.

## Immunoprecipitation and Western blotting

Cells were lysed in NP-40 lysis buffer (50 mM Tris-HCl pH 7.4, 150 mM NaCl, 0.1% Nonidet P-40, 5 mM EDTA, 50 mM NaF, 1 mM Na$_3$VO$_4$, 10% Glycerol) supplemented with protease inhibitor mix (Sigma). For immunoprecipitation, anti-Flag M2 Affinity Gel (Sigma, A2220) was incubated with lysates containing 2 mg of proteins with rotation and washed five times with lysis buffer. Precipitated proteins were eluted by boiling the beads in Laemmli sample buffer. For Western blotting, lysates containing 30 μg of protein or immunoprecipitated proteins were separated by SDS-PAGE, transferred to nitrocellulose membranes, and probed with specific antibodies. Antibodies used for Western blotting: rabbit anti-FAM111A (Abcam, ab184572, 1:1,000); mouse anti-β-actin [clone AC-74] (Sigma, A5316, 1:5,000), rabbit anti-Flag (Cell Signaling, #2368, 1:1,000); rabbit anti-HA (Santa Cruz, sc-805, 1:1,000). Blots were imaged using ChemiDoc MP (Bio-Rad) and analyzed using the Image Lab software (version 6.0.1, Bio-Rad).

## Immunofluorescence and microscopy

For FAM111A staining, cells underwent chromatin pre-extraction with CSK buffer supplemented with Triton X-100 (10 mM HEPES-KOH pH 7.4, 300 mM sucrose, 100 mM NaCl, 3 mM MgCl$_2$, and 0.5% Triton X-100) for 10 min at room temperature. Cells were fixed on coverslips using cold Methanol: Acetone (3:1) for 10 min at −20 °C, washed with PBS for 5 min, then blocked in 1% BSA, 22.52 mg/mL glycine and 0.1% Tween-20 in PBS for 1 hour at room temperature. Following blocking, samples were incubated with primary rabbit anti-FAM111A antibody (Abcam, ab184572, 1:100) in the blocking solution for 2 h at 37 °C. Samples were washed in PBS three times then incubated with secondary antibody, goat anti-rabbit IgG Alexa Fluor 568 (Invitrogen, A-11036, 1:2,000) in blocking solution for 45 min at room temperature. Nuclei were stained with DAPI in PBS and coverslips were mounted with ProLong Gold Glass Antifade Mountant (Thermo Fisher, P36980).

Images were captured using a Nikon SoRa Spinning Disk microscope with the NIS Elements software, and single z slices are shown.

TOP1cc staining was performed as described previously[59]. Briefly, the cells were fixed for 15 min at 4 °C using 4% paraformaldehyde, permeabilized with 0.25% Triton X-100 in PBS for 15 min at 4 °C and treated with 1% SDS in PBS for 5 min at room temperature. The slides were washed five times with wash buffer (0.1% Triton X-100, 0.1% BSA in PBS) and blocked with 10% milk in 10 mM Tris-HCl pH 7.4, 150 mM NaCl. TOP1cc foci were detected with a primary mouse anti-TOP1cc antibody (a gift from Scott Kaufmann, 1:200) and secondary antibody, goat anti-mouse IgG Alexa Fluor 488 (Invitrogen, A-11029, 1:1,000). All images were captured using a Nikon SoRa Spinning Disk microscope, and max intensity projection was generated using Fiji (version 2.9.0)[60]. TOP1cc foci were scored using an ImageJ script on images captured with a 63× objective by a blinded observer.

## DNA combing assay

The DNA combing assay was performed as previously described[61] with some modifications. Briefly, cells were sequentially labeled with 25 μM CldU (Sigma, C6891) for 30 min and 100 μM IdU (Sigma, I7125) for 30 min, followed by a wash with ice-cold PBS to inhibit DNA replication. The IdU labeling was performed in the absence or presence of 30 nM CPT. Cells were trypsinized and resuspended in PBS. Cells ($1 \times 10^5$) were then embedded in a low-melting-point agarose plug. The plugs were incubated for 18 h at 50 °C with 1 mg/ml proteinase K in cell lysis buffer (10 mM Tris-HCl pH 8.0, 1% N-Laroylsarcosine sodium salt, 100 mM EDTA) and then washed three times for 1 h each with TE buffer. The plugs were melted in 0.1 M MES pH 6.5 for 20 min at 70 °C and were incubated for 10 min at 42 °C before β-agarase (NEB, M0392S) was added. After 18 h of β-agarase digestion at 42 °C, DNA solutions were poured into a Teflon reservoir and stretched using an in-house combing machine onto salinized coverslips (Genomic Vision, COV-002-RUO). Combed DNA was baked in an oven for 2 h at 60 °C and denatured in 0.5 M NaOH for 20 min, followed by washing with PBS five times for 1 min each. Coverslips were blocked in PBS with 5% BSA for 10 min at room temperature, then incubated with rat anti-CldU antibody [clone BU1/75 (ICR1)] (Abcam, ab6326, 1:100) and mouse anti-IdU antibody [clone B44] (BD, 347580, 1:20) in PBS with 5% BSA overnight at 4 °C in a wet chamber. After washing coverslips with PBS-T (0.05% Triton X-100) three times for 5 min each on the shaker, coverslips were incubated with Cy5-labeled goat anti-rat IgG (Abcam, ab6565, 1:100) and Cy3-labeled goat anti-mouse IgG (Abcam, ab97035, 1:100) in PBS with 5% BSA for 1 h at room temperature in a wet chamber. Coverslips were washed with PBS-T (0.05% Triton X-100) three times for 5 min each on the shaker and incubated with mouse anti-single strand DNA [clone 16-19] (Millipore, MAB3034, 1:200) in PBS with 5% BSA for 1 h at room temperature in a wet chamber. After washing with PBS-T (0.05% Triton X-100) three times for 5 min each on the shaker, coverslips were incubated with Brilliant Violet 480-labeled goat anti-mouse IgG (Jackson ImmunoResearch, 115-685-166, 1:50) in PBS with 5% BSA for 45 min at room temperature in a wet chamber. After washing with PBS-T (0.05% Triton X-100) three times for 5 min each on the shaker, coverslips were rinsed with water briefly and dehydrated sequentially in 70%, 90%, and 100% ethanol for 2 min at each concentration and air-dried at room temperature. Coverslips were attached to sample holders (Genomic Vision, HLD-001) and scanned with a FiberVision automated fluorescence microscope (Genomic Vision, SCN-001). The length of CldU and IdU tracts were measured by blinded observers using FiberStudio (version 3.2.7, Genomic Vision).

## Multiple sequence alignment (MSA)

MSA of *Homo sapiens* FAM111A SPD with FAM111A orthologs or *Bos taurus* chymotrypsin was generated using Clustal Omega (version 1.2.4)[62,63]. FAM111A SPD amino acid sequences used in alignment include:

*Homo sapiens* FAM111A (Human; UniProt ID: Q96PZ2; 335-611), *Mus musculus* FAM111A (Mouse; UniProt ID: Q9D2L9; 332-613), *Pantherophis guttatus* FAM111A (Corn snake; UniProt ID: A0A6P9C8G8; 368-671), and *Danio rerio* FAM111A-like (Zebrafish; UniProt ID: A0A8M2BD64; 394-686). *Bos taurus* chymotrypsin (XP_003587247) was used in the pairwise sequence alignment with *Homo sapiens* FAM111A SPD.

## Statistics

All graphs were created in GraphPad Prism (version 9.5.0). Statistical significance was determined by a two-tailed unpaired t-test and adjusted for multiple sample comparison using the Holm method in R (version 4.1.2). Slopes for protease assays were calculated using MARS (version 3.41) (BMG Labtech) or GraphPad Prism (version 9.5.0). Statistics for X-ray diffraction experiments are contained in Supplementary Table 2.

## Reporting summary

Further information on research design is available in the Nature Portfolio Reporting Summary linked to this article.

## Data availability

Atomic coordinates and structure factors have been deposited in the PDB under accession numbers 8S9K (SPD S541A) and 8S9L (mini-SPD). The PDB ID for the structure of chymotrypsin co-crystallized with a TFMK transition state analog is 7GCH. The configuration files for the MD simulations are available at Figshare [https://doi.org/10.6084/m9.figshare.24915123]. Source data are provided with this paper.

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

## Acknowledgements

We thank Dr. Scott H. Kaufmann at Mayo Clinic for his generous gift of anti-TOP1cc antibody, and Drs. Di Wu and Grzegorz Piszczek at NHLBI Biophysics Core Facility for the assistance on mass photometry. Confocal imaging was performed at the CCR Microscopy Core Facility at the National Cancer Institute. This study was supported by the National Institutes of Health (R01 CA233700 to Y.J.M. and M.J.S.), the Intramural Research Program of the NIH, National Cancer Institute, Center for Cancer Research (ZIA BC 012086 to Y.J.M.), and Mayo Clinic Startup funds to M.J.S. G.M. acknowledges support from NIH award R35 GM136262. This work is based upon research conducted at the Northeastern Collaborative Access Team beamlines, which are funded by the National Institute of General Medical Sciences from the National Institutes of Health (P30 GM124165) and NIH-ORIP HEI grant (S10OD021527). This research used resources of the Advanced Photon Source, a U.S. Department of Energy (DOE) Office of Science User Facility operated for the DOE Office of Science by Argonne National Laboratory under Contract No. DE-AC02-06CH11357.

## Author contributions

Conceptualization, M.J.S. and Y.J.M.; Investigation, S.P., Y.M., J.R.A., V.M., A.L.W., G.C., B.B., M.V.B., M.J.S., and Y.J.M.; Technical Support, A.T.Q.C; Writing – Original Draft, S.P., J.R.A., Y.M., G.C., B.B., A.L.W., M.J.S., and Y.J.M.; Writing – Review & Editing, S.P., Y.M., J.R.A., A.L.W., V.M., G.M., M.J.S., and Y.J.M.; Funding Acquisition, Y.J.M.; Supervision, G.M., M.J.S., and Y.J.M.

## Funding

## Competing interests

The authors declare no competing interests.

## Additional information

[1]Mayo Clinic Graduate School of Biomedical Sciences, Mayo Clinic, Rochester, MN, USA. [2]Department of Oncology, Division of Oncology Research, Mayo Clinic, Rochester, MN, USA. [3]Developmental Therapeutics Branch, Center for Cancer Research, National Cancer Institute, Bethesda, MD, USA. [4]Department of Biochemistry and Molecular Biology, Mayo Clinic, Rochester, MN, USA. [5]Department of Molecular Pharmacology and Experimental Therapeutics, Mayo Clinic, Rochester, MN, USA. [6]These authors contributed equally: Sowmiya Palani, Yuka Machida. [7]These authors jointly supervised this work: Matthew J. Schellenberg, Yuichi J. Machida. ✉e-mail: schellenberg.matthew@mayo.edu; yuichi.machida@nih.gov

