## [Peer Review File · Nature Communications]

Dimerization-dependent serine protease activity of FAM111A prevents replication fork stalling at topoisomerase 1 cleavage complexesReviewer #1 (Remarks to the Author):

In this manuscript, Machida and co-workers claimed to study the effect of dimerization on FAM111A serine protease activity and also on its allosteric property and biological functions such as antiviral activity and DNA replication using biochemical and structural biology tools.

The comments are as follows:

The title 'FAM111A is a dimerization-dependent serine protease' is a very non-descript one that does not bring out the essence of the work and needs revision.

The manuscript studies the role of dimerization of SPD of FAM111A on its activity and its associated biological functions. To look into this, they cloned SPD (a part of the full-length protein) and a few of its mutants and performed in vitro protease activity as well as studied their oligomeric properties. They also looked into the crystal structure of the SPD and a smaller mini SPD domain.

However, there are serious concerns regarding the experimental designs and subsequent interpretation of data:

1. SPD is a part of the full-length protein, therefore the oligomeric property and stability related to its oligomerization might not be relevant in the biological context. For example, one mutant they thought would make it a monomer, still showed dimeric property-and the reason provided that the concentration is high, is not clear to me.

2. Oligomeric property was determined using SEC and AUC (sedimentation velocity). However, the oligomeric property is best determined through sedimentation equilibrium studies. Moreover, no K_d (dimerization constant) value has been provided that could have explained the strength of the dimers in all the protein variants and could have unambiguously explained the reason for the dimeric property observed in the mutant. Therefore, the data provided is very qualitative, not quantitative as expected from AUC. Since the entire manuscript depends on the oligomeric property, it makes no sense to not do so as it weakens the data provided and the interpretations made.

3. Many mutants were made that they predicted to make the SPD monomer-however, no rationale, however, has been provided on how those residues were chosen other than just structure-based wild guesses. An in silico analysis would have been a more organized approach.

4. X-ray crystallography was performed using a very high concentration of SPD domain that showed it is a dimer. Moreover, the monomer also showcased dimeric property and hence they made a truncated version of SPD (mini SPD) to determine its oligomeric property. Again, the entire experimental design is quite questionable. A dimer might be formed due to contacts at different parts of the protein, therefore chopping off a major part of the protein and concluding it is a monomer makes no sense; neither it proves anything nor it represents what actually happens in the cellular milieu.

5. 'The differences observed between the two chains of the mini-SPD structure and the dimeric SPD as well as the higher B-factors observed for residues 536-538 encompassing the oxyanion hole and S1 pocket indicate that the monomeric SPD is more disordered. This suggests that dimerization of the SPD is associated with a disorder-to-order transition that stabilizes the oxyanion hole residues in the conformation that is competent for catalysis'

This particular interpretation is an overstatement and is not backed by sufficient data. In fact, there is hardly any solid data to support it. Although some enzyme studies have been done, no data analysis using the Michaelis Menten equation has been made and no MM parameters have been provided that would show an allosteric property (Hill coefficient, V_{max} , K_m , etc). I am not sure on what basis the authors would say that the oxyanion hole is disrupted. Crystallography should have been backed with enzyme kinetics data.

6. 'This might suggest that Tyr414 plays an additional role in the activation mechanism, possibly by transmitting the allosteric changes to other parts of the SPD such as the loop L6'

Again, this statement is not backed by any data confirming the same.

Allostery is a property that has not been proven anywhere in the manuscript; moreover, the full-length protein might have provided a more comprehensive picture of the same. Therefore, cell-based studies would have been helpful since full length protein could not be purified in the bacterial system.

In the discussion, the authors talk about making allosteric inhibitors, which is more like talking in the air with no solid basis whatsoever.

7. SPD R569H and D528G measured using Suc-AAP. However, why the activity of monomeric- = R569H is higher has not been explained. It was very much needed since it is counterintuitive and brings to question what is actually tried to be conveyed by this work.

8. 'Expression of WT FAM111A prevented the spontaneous formation of TOP1cc foci in FAM111A KO, as described previously. In contrast, the expression of the monomeric mutants failed to prevent TOP1cc accumulation (Fig. 6b,c), despite their comparable expression levels to that of WT (Fig. 6a)'

These are full-length FAM111A protein variants used in cell-based assays and not the SPD whose oligomeric property was tested in vitro. Therefore, how it has been confirmed that the full-length mutants are also monomeric?

9. How Functional assay correlates with in vitro protease activity? Since FAM111A localizes at replication forks and promotes DNA replication at protein obstacles through its protease activity- this could have been tested using appropriate cell-based assays to directly link protease activity with cellular functions.

OVERALL: The work is very qualitative with data incomplete and hence it fails to prove the point addressed in the manuscript.

Reviewer #2 (Remarks to the Author):

In this study, the authors investigate the molecular architecture of the serine protease FAM111A, which is important for DNA replication in the presence of covalent DNA-protein crosslinks (DPCs). Missense germline mutations of FAM111A residues are also associated with genetic developmental disorders. The authors propose that FAM111A is a dimerization-dependent protease. X-ray crystal structures reveal that the SPD dimerizes via an N-terminal helix, inducing an allosteric activation cascade from the dimerization sensor loop to the oxyanion hole through disorder-to-order transitions. Replacement of key residues in FAM111A's dimerization interface results in loss or reduction of protease activity and protein stability in vitro, suggesting that enzyme oligomerization is important for FAM111A function. Moreover, dimerization-defective FAM111A variants fail to promote replication at DPC obstacles in cells.

Overall, these discoveries provide important information about FAM111A regulation. The manuscript is well-written, and the experiments support the conclusions. The following comments to the authors could guide some modest revisions.

1. The authors propose dimerization as a crucial regulatory mechanism that safeguards against uncontrolled FAM111A proteolysis. This implies that dimerization may be regulated. In the present study, the authors only investigate the catalytic SPD domain but it would be important to test whether the full-length protein also dimerizes and whether dimerization is affected by autocatalytic cleavage. Along these lines, autocleavage of dimerization-defective FAM111A V347D and V351D variants should be assessed in cells, as the authors have done in previous work. Could the increased expression levels of these variants in cells be related to defective autocleavage? FAM111A disease variants are generally believed to cause hyperactive protease activity. Interestingly, one mutation tested in the manuscript in context of the catalytic SPD domain decreased while another one increased activity. This may suggest that different patient mutations

can have different effects on autocleavage than on substrate cleavage. Thus, it would be worthwhile to test whether the reported increased autocleavage of patient variants in cells requires dimerization of the SPD domain or not.

2. The miniSPD and dimer-interface mutant variants are not only defective in dimerization and activity but are also less stable in vitro. While this suggests that dimerization is important for FAM111A, a structural biology expert should comment on whether some of the conclusions on allosteric activation of the active site are entirely justified or could be biased by unspecific folding defects.

3. The authors conclude "...that the FAM111A SPD active-site architecture is tailored to cleave smaller substrates, such as linker regions or disordered substrates, and not globular proteins." However, it is also conceivable that FAM111A substrates are actively unfolded prior to proteolysis, as recently shown for SPRTN substrates.

4. Figure 3C: The colors used are difficult to distinguish.

5. Page 9: "To disrupt the β 10 interface, we generated a Thr563 to proline (P) (T563P) mutant, which would disrupt the β 10 sheet hydrogen bonds (Supplementary Fig. 3a). We generated recombinant SPD containing these mutations (Supplementary Fig. 3b)..." Neither Fig. S3A nor S3B shows data on the T563P variant.

Reviewer #3 (Remarks to the Author):

FAM111A is a protein-coding gene important for cell-cycle regulation, and nuclear localization. The C-terminal part of the protein shares homology with trypsin-like peptidases and it contains a PCNA-interacting peptide (PIP) box, that is necessary for its co-localization with proliferating cell nuclear antigen (PCNA). In this study by Palani et al., investigated the structural and functional basis of the C-terminal part of the FAM111A. They have reported the first crystal structure of the C-terminal FAM111A (However, AlphaFold model of full-length FAM111A (AF-Q96PZ2-F1) is already solved at a high confidence level) as both dimeric (active) and monomeric (inactive) configurations. They find that this protein exists as a dimer in solution and is necessary for its function whereas monomeric protein did not show any activity. Site-directed mutagenesis confirms the important residues for protein dimerization. Their structural data is supported by in-vitro protease activity data as well as cellular functional data to show that dimeric FAM111A is critical for TOP1cc accumulation and replication fork stalling in cells. Thus, the manuscript follows a previous publication from the same group (Kojima et al., 2020) where they show the different direction of the FAM111A, a PCNA-interacting protein, which plays an important role in mitigating the effect of protein obstacles on replication forks.

This is a well-designed study with a substantial amount of high-quality data. That FAM111A exists as a dimer in solution and is required for its function are significant findings, providing novel insights into how serine proteases are structurally flexible in their function. Although the work appears rigorous and essentially descriptive, the structural aspects do not break any significant new ground. Thus, I feel this report is too preliminary for publication in Nature Communication.

The following comments are provided for improving the manuscript.

1. Authors claim that FAM111A has chymotrypsin-like protease activity. However, they have just shown that FAM111A could cleave phenylalanine (Phe/F) at the P1 position. In contrast, chymotrypsin cleaves after Phe/Tyr/Trp residues. How could it be valid for P1 substitution by Tyr/Y or Trp/W? FAM111A still be able to cleave the substrate?

2. I would describe the percentage of amino acid sequence identity and RMSD difference between FAM111A SPD and chymotrypsin in the result section. This will give the readers some idea of the difference between FAM111A SPD and previously characterized chymotrypsin.

3. The crystal structures of C-terminal FAM111A are solved by molecular replacements using the Alphafold model. However, the authors did not report how good the alpha fold model to their crystal structure. I would add TFZ values and LLG values resulting from molecular replacement run into the method section. This will give a better understanding to the reader.

4. The reported final Rfree value for the Mini SPD crystal structure is significantly high for 1.85A data. It is important to acknowledge what drives this high R-free value.

5. Page 8, second paragraph- last sentence: Report refers to Figures 3a and 3b to show a network of salt bridges and H-bonds. I can't see such interactions shown in that figure. also, this follows by, Page 9, line 2: can't see T563P mutant shown in supplementary figure 3a

I would list all these key interactions in the figure or in a separate panel. It makes easier for the reader to follow the idea.

6. Page 09- paragraph 1, last sentence: "...two sets of 6 hydrogen bonds and three salt bridges...". I would suggest authors to list those interactions in a figure or list them in-text.

7. An interesting point is that both SPD mutants, V351 and V347D/V351D still retained the dimer interface. It is unclear what other residues/interactions are driving this dimer formation.

8. How similar is the structure of mini-SPD to wt-SPD? I would report the RMSD value and any noticeable local structural changes/loop movements here. Also, this can be compared with the Alphafold structure available and discuss any structural changes.

9. Few missing abbreviations, please use the full term when you use it the first time.

- SV40 and PCNA

10. Missing reference on page 4, paragraph 2, line 5: "but FAM111A is one of the few proteases in this family that mainly localizes in the nucleus [REF]"

Reviewer Comments and Author Responses

Reviewer #1 (Remarks to the Author):

In this manuscript, Machida and co-workers claimed to study the effect of dimerization on FAM111A serine protease activity and also on its allosteric property and biological functions such as antiviral activity and DNA replication using biochemical and structural biology tools.

The comments are as follows:

The title 'FAM111A is a dimerization-dependent serine protease' is a very non-descript one that does not bring out the essence of the work and needs revision.

Author response: The title was revised to a more descriptive one to convey the main point of the study.

The manuscript studies the role of dimerization of SPD of FAM111A on its activity and its associated biological functions. To look into this, they cloned SPD (a part of the full-length protein) and a few of its mutants and performed in vitro protease activity as well as studied their oligomeric properties. They also looked into the crystal structure of the SPD and a smaller mini SPD domain.

However, there are serious concerns regarding the experimental designs and subsequent interpretation of data:

1. SPD is a part of the full-length protein, therefore the oligomeric property and stability related to its oligomerization might not be relevant in the biological context. For example, one mutant they thought would make it a monomer, still showed dimeric property-and the reason provided that the concentration is high, is not clear to me.

Author response: We agree with the reviewer that it is important to assess whether the mutations we engineered to disrupt SPD dimerization also alter dimerization of the full-length protein. In the revised manuscript, we have addressed this point by three new experiments.

- 1) **Size-exclusion chromatography (SEC) of recombinant full-length FAM111A proteins.** We produced Strep-tagged full-length FAM111A S541A proteins, both with and without the V347D mutation, in insect cells and purified using the Strep tag (Fig. 1b and Supplementary Fig. 4a). Using size-exclusion chromatography, we show that Strep-FAM111A S541A protein eluted at a volume corresponding to an estimated molecular weight of 199 kDa, roughly matching the calculated molecular weight of the Strep-FAM111A dimer (Fig. 3g). In contrast, FAM111A S541A/V347D eluted at a volume corresponding to an estimated molecular weight of 96 kDa, close to the expected elution volume for the monomeric Strep-FAM111A protein.
- 2) **Mass photometry of full-length FAM111A (Fig. 3h).** We note that size determination by SEC can be affected by the shape of the protein. Thus, we performed experiments using

mass photometry, a technique that measures the mass of individual biomolecules by analyzing their light-scattering properties in a shape-independent manner. The results suggested that full-length FAM111A (S541A) largely exists in a dimer form (145 KDa), and the V347D mutant is mostly monomeric (84 KDa, Fig. 3h).

- 3) **Coimmunoprecipitation of FAM111A-HA with FAM111A-Flag co-expressed in human cells.** We coexpressed FAM111A-HA and FAM111A-Flag in 293T cells and examined their interaction. Our results showed that the V347D mutation, and to a lesser extent the V351D mutation, abrogated intermolecular interaction of the full-length FAM111A (Supplementary Fig. 6c).

Based on these findings, we conclude that full-length FAM111A also exists in a dimeric form, and the V347D mutation disrupts this dimerization.

The V347D and V351D mutants were engineered to disrupt the molecular interactions at the dimer interface. The consequence of such mutations (which both would disrupt coiled-coil hydrophobic interactions and introduce charge-charge repulsion) is to destabilize the dimeric state which effectively raises the K_d of the dimer. Consequently, by Le Chatelier's principle, if the concentration of the mutant monomeric protein increases and approaches the K_d of the mutant protein, the system will be in a dynamic equilibrium with the dimeric state. Please bear in mind that the estimated concentration of protein in the SPD and min-SPD crystals is ~ 16 mM (!) driven by the precipitants used in the vapor diffusion crystallization experiments. At these very high concentrations (which are necessary for crystals to form), the $\alpha 1$ dimer interface is still observed with the mutants (except for min-SPD), which prevented us from studying monomeric structures by X-ray crystallography. We have modified the text of the manuscript to clarify this.

2. Oligomeric property was determined using SEC and AUC (sedimentation velocity). However, the oligomeric property is best determined through sedimentation equilibrium studies. Moreover, no K_d (dimerization constant) value has been provided that could have explained the strength of the dimers in all the protein variants and could have unambiguously explained the reason for the dimeric property observed in the mutant. Therefore, the data provided is very qualitative, not quantitative as expected from AUC. Since the entire manuscript depends on the oligomeric property, it makes no sense to not do so as it weakens the data provided and the interpretations made.

Author response: Although we agree with the reviewer that sedimentation equilibrium studies are suited for studying oligomeric properties, the K_d for FAM111A SPD is far below the concentrations needed to detect the sample in such analyses, and therefore it is not a suitable technique to determine K_d in the low nanomolar range. Instead, we have analyzed the dimerization property of full-length FAM111A and determined its dimerization K_d using mass photometry. The acquired data suggest that full-length FAM111A exists in a dimer form with estimated dimerization K_d as 4.8 nM, which is consistent with the estimated K_d of < 49 nM obtained from SPD enzyme activity (Supplementary Fig. 1a). We were not able to estimate the K_d values of the full-length monomeric mutant (FAM111A FL V347D) using mass photometry due to the insufficient amount of the dimer population with this mutant. We believe that these

quantitative data on dimerization of full-length FAM111A strengthened our conclusion that dimerization is important for the FAM111A protease activity. We note that it would be ideal to also be able to analyze the K_d of the SPD domain alone, however, the monomeric SPD is below the size limit (40 kDa) that can be accurately counted on our instrument. We thank the reviewer for the constructive comments.

3. Many mutants were made that they predicted to make the SPD monomer-however, no rationale, however, has been provided on how those residues were chosen other than just structure-based wild guesses. An in silico analysis would have been a more organized approach.

Author response: We apologize for not thoroughly explaining how we designed the dimer interface mutations. In this study, we have used the experimentally determined atomic coordinates to rationally design the point mutations V347D, V351D, and T563P to disrupt potential dimer interfaces. We selected these residues for mutagenesis because they form inter-molecular interactions, but do not contribute to the internal structure of the protein. When evaluating potential effects of mutagenesis, we indeed used in silico analyses and modeled the common rotamers to ensure such mutations cannot be accommodated in the existing interface by any rotamer. In the case of the α 1 helix interface, we chose to mutate the two valine residues to aspartate to both disrupt the hydrophobic coiled-coil interactions between the α 1 helices and introduce charge-repulsion to destabilize the interface and raise the dimer K_d far above physiological concentrations. We chose the T563P mutation since the proline side chain occludes the hydrogen bonds that are observed between adjacent β 10 strands as seen in the crystal structure. We now include updated figure panels 3c,d to highlight the steric clashes that would be generated by these mutants upon dimer formation.

4. X-ray crystallography was performed using a very high concentration of SPD domain that showed it is a dimer. Moreover, the monomer also showcased dimeric property and hence they made a truncated version of SPD (mini SPD) to determine its oligomeric property. Again, the entire experimental design is quite questionable. A dimer might be formed due to contacts at different parts of the protein, therefore chopping off a major part of the protein and concluding it is a monomer makes no sense; neither it proves anything nor it represents what actually happens in the cellular milieu.

Author response: In the revised manuscript, we have successfully produced and analyzed full-length FAM111A proteins with various mutations. Our new experiments using size-exclusion chromatography, mass photometry, and coimmunoprecipitation demonstrated that dimerization of full-length FAM111A is disrupted by the V347D mutation. These data suggest that the α 1 helix within the SPD is an important interface for the dimerization in full-length FAM111A.

5. The differences observed between the two chains of the mini-SPD structure and the dimeric SPD as well as the higher B-factors observed for residues 536-538 encompassing the oxyanion hole and S1 pocket indicate that the monomeric SPD is more disordered. This suggests that dimerization of the SPD is associated with a disorder-to-order transition that stabilizes the

oxyanion hole residues in the conformation that is competent for catalysis'

This particular interpretation is an overstatement and is not backed by sufficient data. In fact, there is hardly any solid data to support it. Although some enzyme studies have been done, no data analysis using the Michaelis Menten equation has been made and no MM parameters have been provided that would show an allosteric property (Hill coefficient, V_{max} , K_m , etc). I am not sure on what basis the authors would say that the oxyanion hole is disrupted. Crystallography should have been backed with enzyme kinetics data.

Author response: Our conclusion that the oxyanion hole is disordered in the monomeric form is based on direct observations of the crystal structure of the wildtype and monomeric mutant SPD proteins, and supported by MD simulations and enzyme assays that link the monomeric state to the disordered oxyanion hole and inactive enzyme. Similar approaches were used previously with other dimer proteases from human herpesvirus to conclude that the dimerization-induced activation is associated with structural changes in the oxyanion hole (Ref. #37 and #38).

Determining MM parameters was challenging due to the suboptimal substrate with high K_m , and we were unable to measure V_{max} . Furthermore, we cannot assay the monomeric double V/D mutant or mini-SPD as they lack any detectable activity. Although we do note that the term allostery is used in Refs 37/38 to describe similar dimerization-mediated active site conformational changes, we agree with the reviewer that we do not have sufficient data to fully support the allosteric property of dimerization-dependent FAM111A activity. Therefore, the notion of allostery has been removed in the revised manuscript.

6. 'This might suggest that Tyr414 plays an additional role in the activation mechanism, possibly by transmitting the allosteric changes to other parts of the SPD such as the loop L6'

Again, this statement is not backed by any data confirming the same.

Allostery is a property that has not been proven anywhere in the manuscript; moreover, the full-length protein might have provided a more comprehensive picture of the same. Therefore, cell-based studies would have been helpful since full length protein could not be purified in the bacterial system.

Author response: Please also see our response to point #5 above. We have modified the sentence as follows: "This might suggest that Tyr414 plays an additional role in the activation cascade."

In the discussion, the authors talk about making allosteric inhibitors, which is more like talking in the air with no solid basis whatsoever.

Author response: Our intention of this discussion was to explore development of dimerization blockers to inhibit FAM111A. Similar idea has been pursued for developing inhibitors for dimer proteases from human herpesvirus (Ref. #38). We have removed the word "allosteric" from the

sentence in the revised manuscript to reflect our intention more precisely (page 20, first paragraph). Please also see our response to point #5 above.

7. SPD R569H and D528G measured using Suc-AAP. However, why the activity of monomeric R569H is higher has not been explained. It was very much needed since it is counterintuitive and brings to question what is actually tried to be conveyed by this work.

Author response: The data presented in the original Fig. S1 included the R569H mutant but not the monomeric R569H mutant. In the revised manuscript, we include the monomeric R569H mutants (SPD R569H/V347D and R569H/V351D), showing that the mutations in the dimerization interface diminished the activity of the R569H patient mutant (Fig. 6b). This data indicates that substrate cleavage by the patient mutant is still dependent on dimerization like WT, suggesting that the patient-associated mutation did not bypass the requirement of enzyme dimerization.

8. 'Expression of WT FAM111A prevented the spontaneous formation of TOP1cc foci in FAM111A KO, as described previously. In contrast, the expression of the monomeric mutants failed to prevent TOP1cc accumulation (Fig. 6b,c), despite their comparable expression levels to that of WT (Fig. 6a)'

These are full-length FAM111A protein variants used in cell-based assays and not the SPD whose oligomeric property was tested *in vitro*. Therefore, how it has been confirmed that the full-length mutants are also monomeric?

Author response: In the revised manuscript, we have examined the oligomeric property of full-length FAM111A *in vivo* using co-immunoprecipitation. Our data indicated that the inter-molecular interaction between FAM111A-Flag and FAM111A-HA was markedly diminished when the V347D mutations were introduced to both proteins. We believe that the new data strengthen our conclusion that dimerization is important for the *in vivo* functions of FAM111A.

9. How Functional assay correlates with *in vitro* protease activity? Since FAM111A localizes at replication forks and promotes DNA replication at protein obstacles through its protease activity-this could have been tested using appropriate cell-based assays to directly link protease activity with cellular functions.

Author response: We agree with the reviewer that it is important to test how functional assays *in vivo* correlate with protease activity *in vitro*. We have reported the importance of FAM111A activity in our previous paper (Ref. #10, Kojima et al., Nature Commun., 2020) using the active site mutation S541A. The importance of dimerization was tested using V347D and V351D mutants (originally in Fig. 6, but now in Fig. 7c,d,h).

In the current revision, we have included new data showing that the K348A mutation diminished the function of FAM111A in reducing TOP1ccs levels (Fig. 7e,f) and promoting DNA replication at CPT-induced TOP1ccs *in vivo* (Fig. 7i). These data suggest that Lys348 that makes a contact with the dimerization sensor loop (Fig. 4e) is important for the *in vivo* function of

FAM111A, which is consistent with our model that the Lys348 residue is crucial for FAM111A activity.

OVERALL: The work is very qualitative with data incomplete and hence it fails to prove the point addressed in the manuscript.

Author response: We believe that the new data regarding the dissociation constants of full-length FAM111A have significantly improved the quantitative aspect of our study. Additionally, the new experiments using full-length FAM111A provides more complete data on FAM111A dimerization and activity, as well as the malfunction of FAM111A caused by patient mutations. We thank the reviewer for the valuable comments.

Reviewer #2 (Remarks to the Author):

In this study, the authors investigate the molecular architecture of the serine protease FAM111A, which is important for DNA replication in the presence of covalent DNA-protein crosslinks (DPCs). Missense germline mutations of FAM111A residues are also associated with genetic developmental disorders. The authors propose that FAM111A is a dimerization-dependent protease. X-ray crystal structures reveal that the SPD dimerizes via an N-terminal helix, inducing an allosteric activation cascade from the dimerization sensor loop to the oxyanion hole through disorder-to-order transitions. Replacement of key residues in FAM111A's dimerization interface results in loss or reduction of protease activity and protein stability in vitro, suggesting that enzyme oligomerization is important for FAM111A function. Moreover, dimerization-defective FAM111A variants fail to promote replication at DPC obstacles in cells.

Overall, these discoveries provide important information about FAM111A regulation. The manuscript is well-written, and the experiments support the conclusions. The following comments to the authors could guide some modest revisions.

Author comments: We would like to thank the reviewer for the supportive comments.

1. The authors propose dimerization as a crucial regulatory mechanism that safeguards against uncontrolled FAM111A proteolysis. This implies that dimerization may be regulated. In the present study, the authors only investigate the catalytic SPD domain but it would be important to test whether the full-length protein also dimerizes and whether dimerization is affected by autocatalytic cleavage.

Author response: We agree with the reviewer that it is important to test our findings from the study on SPD in the full-length protein context. In the revised manuscript, we have addressed this point by three new experiments.

- 1) **Size-exclusion chromatography (SEC) of recombinant full-length FAM111A proteins.**
We produced Strep-tagged full-length FAM111A S541A proteins, both with and without

the V347D mutation, in insect cells and purified using the Strep tag (Fig. 1b and Supplementary Fig. 4a). Using size-exclusion chromatography, we show that Strep-FAM111A S541A protein eluted at a volume corresponding to an estimated molecular weight of 199 kDa, roughly matching the calculated molecular weight of the Strep-FAM111A dimer (Fig. 3g). In contrast, FAM111A S541A/V347D eluted at a volume corresponding to an estimated molecular weight of 96 kDa, close to the expected elution volume for the monomeric Strep-FAM111A protein.

- 2) **Mass photometry of full-length FAM111A (Fig. 3h).** We note that size determination by SEC can be affected by the shape of the protein. Thus, we performed experiments using mass photometry, a technique that measures the mass of individual biomolecules by analyzing their light-scattering properties in a shape-independent manner. The results suggested that full-length FAM111A (S541A) largely exists in a dimer form (145 KDa), and the V347D mutant is mostly monomeric (84 KDa, Fig. 3h).
- 3) **Coimmunoprecipitation of FAM111A-HA with FAM111A-Flag co-expressed in human cells.** We coexpressed FAM111A-HA and FAM111A-Flag in 293T cells and examined their interaction. Our results showed that the V347D mutation, and to a lesser extent the V351D mutation, abrogated intermolecular interaction of the full-length FAM111A (Supplementary Fig. 6c).

Based on these findings, we conclude that full-length FAM111A also exists in a dimeric form, and the V347D mutation disrupts this dimerization.

Regarding the effect of autocleavage on dimerization, we believe that autocleavage at Phe334 does not alter the dimerization status of the SPD, as the α 1 helix dimer interface is downstream of the autocleavage site. Since autocleavage would release active SPD dimers, it is possible that autocleavage activity elevated in patient mutants might indirectly cause FAM111A hyperactivity. This notion is now discussed in Page 19, line 14-17.

Along these lines, autocleavage of dimerization-defective FAM111A V347D and V351D variants should be assessed in cells, as the authors have done in previous work.

Author response: As requested by the reviewer, we have examined the autocleavage activity of the dimerization-defective mutant FAM111A V347D, as well as the K348A and Y414A mutants. Our data indicate that these mutations did not have any impact on the autocleavage of wild-type and patient-associated mutants (Fig. 6c and Supplemental Fig. 6d,e). This result is surprising, given that autocleavage activity was previously assumed to reflect substrate-cleaving activity, which is dimerization-dependent as we show in this study. Based on these observations, we have concluded that the autocleavage activity of wild-type and patient-associated mutants is distinct from their substrate cleavage activity. We thank the reviewer for the suggestion, which provided new insight into the FAM111A autocleavage activity. We discussed potential implications of this finding for the patient-associated mutations in the revised manuscript.

Could the increased expression levels of these variants in cells be related to defective autocleavage?

Author response: Since our new data suggests that the V347D mutation does not affect autocleavage activity of FAM111A (Supplementary Fig. 6e), we believe that this difference in expression levels in Fig. 7a is unlikely to be related to autocleavage. Rather, it is more likely due to differences in the toxicities as the V347D mutant has diminished substrate-cleaving activity (Fig. 5b). This notion has been incorporated in the revised manuscript (Page 16, lines 4-6).

FAM111A disease variants are generally believed to cause hyperactive protease activity. Interestingly, one mutation tested in the manuscript in context of the catalytic SPD domain decreased while another one increased activity. This may suggest that different patient mutations can have different effects on autocleavage than on substrate cleavage. Thus, it would be worthwhile to test whether the reported increased autocleavage of patient variants in cells requires dimerization of the SPD domain or not.

Author response: As requested by the reviewer, we have examined the effect of the dimerization-disrupting mutation (V347D) on autocleavage activity of patient-associated mutants (R569H and D528G). While the V347D mutation disrupted oligomerization of full-length FAM111A in coimmunoprecipitation experiments (Supplementary Fig. 6c), it did not affect autocleavage activity of the patient mutants (Fig. 6c and Supplementary Fig. 6d). Based on these data, we conclude that autocleavage activity of the two patient-associated mutants tested in this study do not require dimerization of FAM111A. Since substrate cleavage was slightly elevated in one mutant (R569H) and slightly reduced in another (D528G), we did not observe strong correlation between autocleavage and substrate cleavage activities. As mentioned above, this notion is discussed in more detail in Page 19, second paragraph.

2. The miniSPD and dimer-interface mutant variants are not only defective in dimerization and activity but are also less stable *in vitro*. While this suggests that dimerization is important for FAM111A, a structural biology expert should comment on whether some of the conclusions on allosteric activation of the active site are entirely justified or could be biased by unspecific folding defects.

Author response: In any structure-guided mutagenesis experiment we are very cognizant that unintended impacts on folding can affect experiments. We use several best practices to both prevent the introduction of folding defects, and to identify them when they do occur. The approach used in this manuscript is similar to that used in one of our previous publications (Tumbale *et al.*, EMBO 2018) to characterize the impact of mutations in APTX (in this case they were patient-derived mutations) where we were able to identify and distinguish mutants that caused protein folding defects from those that did not.

Here in this report, we generated dimerization defective mutants by 1) mutating residues (V347 and V351) that do not form part of the protein core and would be on the surface in the monomeric state, 2) we compare the effects of multiple mutations that have the

same effect (the two V/D mutants, double V/D mutant, and mini-SPD), 3) V/D mutants express in *E. coli* and human cells at levels comparable or above that of WT, indicating they fold stably, 4) purify the recombinant proteins via size exclusion chromatography and ion exchange, where they elute as a single peak indicating a uniform folded state, 5) evaluate protein stability by thermal shift assays, 6) obtain crystal structures which show conformational changes are limited to specific regions of the protein, 7) provide supporting MD simulation data that also indicate overall protein folding is not disrupted in silico. The thermal shift assay data (Fig. S5B) is particularly compelling, as dimer disruption by any of the mutants has a similar effect of T_m , and the effects are not additive suggesting the change in T_m is due to solely dimer disruption alone, and not the cumulative effects of core protein destabilization. We also note that the lower T_m of the monomeric mutants ($\sim 50^\circ\text{C}$) is still well above physiological temperatures, and comparable to many other human proteins.

Furthermore, we cite examples of other proteases (Ref #37, 38) that are regulated by disordering of the oxyanion hole that establish precedence for the feasibility of the mechanism proposed in our manuscript.

3. The authors conclude “..that the FAM111A SPD active-site architecture is tailored to cleave smaller substrates, such as linker regions or disordered substrates, and not globular proteins.” However, it is also conceivable that FAM111A substrates are actively unfolded prior to proteolysis, as recently shown for SPRTN substrates.

Author response: The notion that substrates might need to be unfolded prior to proteolysis has been incorporated in the revised manuscript (Page 18, line 25 to Page 19, line 3). We thank the reviewer for the valuable comment.

4. Figure 3C: The colors used are difficult to distinguish.

Author response: The colors of the graph (Fig. 3e in the revised manuscript) have been updated to be more distinct.

5. Page 9: "To disrupt the $\beta 10$ interface, we generated a Thr563 to proline (P) (T563P) mutant, which would disrupt the $\beta 10$ sheet hydrogen bonds (Supplementary Fig. 3a). We generated recombinant SPD containing these mutations (Supplementary Fig. 3b)..." Neither Fig. S3A nor S3B shows data on the T563P variant.

Author response: The figure panels have been corrected to include the sequence alignment around Thr563 (Supplementary Fig. 3b) and SDS-PAGE gel image of the SPD T563P mutant (Supplementary Fig. 4a).

Reviewer #3 (Remarks to the Author):

FAM111A is a protein-coding gene important for cell-cycle regulation, and nuclear localization. The C-terminal part of the protein shares homology with trypsin-like peptidases and it contains

a PCNA-interacting peptide (PIP) box, that is necessary for its co-localization with proliferating cell nuclear antigen (PCNA). In this study by Palani et al., investigated the structural and functional basis of the C-terminal part of the FAM111A. They have reported the first crystal structure of the C-terminal FAM111A (However, AlphaFold model of full-length FAM111A (AF-Q96PZ2-F1) is already solved at a high confidence level) as both dimeric (active) and monomeric (inactive) configurations. They find that this protein exists as a dimer in solution and is necessary for its function whereas monomeric protein did not show any activity. Site-directed mutagenesis confirms the important residues for protein dimerization. Their structural data is supported by in-vitro protease activity data as well as cellular functional data to show that dimeric FAM111A is critical for TOP1cc accumulation and replication fork stalling in cells. Thus, the manuscript follows a previous publication from the same group (Kojima et al., 2020) where they show the different direction of the FAM111A, a PCNA-interacting protein, which plays an important role in mitigating the effect of protein obstacles on replication forks.

This is a well-designed study with a substantial amount of high-quality data. That FAM111A is exist as a dimer in solution and is required for its function are significant findings, providing novel insights into how serine proteases are structurally flexible in their function.

Author comments: We would like to thank the reviewer for the positive comments.

Although the work appears rigorous and essentially descriptive, the structural aspects do not break any significant new ground. Thus, I feel this report is too preliminary for publication in Nature Communication.

Author response: While AlphaFold can predict the overall fold of the monomer, and also dimer structures if prompted, it predicts neither the mechanistic details of the coupled dimerization and order-to-disorder transition, nor links the functional consequences of the oligomeric state on protease activity and cellular DNA repair. Our study addresses this limitation by pinpointing regions showing structural differences between the wild-type and monomeric mutant of SPD. Subsequently, we assess the functional significance of these regions in terms of enzyme activity and in vivo functions. We believe that our study offers valuable insights into how FAM111A dimerization is linked to its activity. While alphafold was useful for producing a molecular replacement search model, none of these advances are interpretable from the alphafold prediction alone.

The following comments are provided for improving the manuscript.

1. Authors claim that FAM111A has chymotrypsin-like protease activity. However, they have just shown that FAM111A could cleave phenylalanine (Phe/F) at the P1 position. In contrast, chymotrypsin cleaves after Phe/Tyr/Trp residues. How could it be valid for P1 substitution by Tyr/Y or Trp/W? FAM111A still be able to cleave the substrate?

Author response: We have expanded our analysis of the P1 specificities of the FAM111A SPD to include substrates with Phe, Tyr, and Trp and compared them with those of chymotrypsin side

by side. As shown in Supplementary Table 1, our data indicated that FAM111A also cleaves at after Tyr and Trp but much less efficiently than after Phe. This trend is similar to chymotrypsin activity although FAM111A exhibits stronger preference for Phe at the P1 residue than chymotrypsin. Therefore, we propose that FAM111A exhibits chymotrypsin-like activity *in vitro*.

2. I would describe the percentage of amino acid sequence identity and RMSD difference between FAM111A SPD and chymotrypsin in the result section. This will give the readers some idea of the difference between FAM111A SPD and previously characterized chymotrypsin.

Author response: We have added a sequence alignment of chymotrypsin and FAM111A SPD in Supplementary Fig. 1b. The sequence identity (16.8%) and the RMSD value (2.2 Å) between the two enzymes are now described in the Result section (Page 7, line 1 and Page 8, line 8).

3. The crystal structures of C-terminal FAM111A are solved by molecular replacements using the Alphafold model. However, the authors did not report how good the alpha fold model to their crystal structure. I would add TFZ values and LLG values resulting from molecular replacement run into the method section. This will give a better understanding to the reader.

Author response: The TFZ and LLG values resulting from the molecular replacement is now reported in the revised manuscript in the methods section: SPD solved with alphafold model = TFZ score of 33.4 and LLG of 1309, mini-SPD solved with SPD model = a TFZ score of 36.3 and LLG of 1285.

4. The reported final Rfree value for the Mini SPD crystal structure is significantly high for 1.85A data. It is important to acknowledge what drives this high R-free value.

Author response: The reviewer is correct to point out that the Rfree metric is higher for the mini-SPD structure. We attribute this to the presence of more mobile residues in and within the vicinity of regions that are part of the disorder-to-order transition. These residues are more challenging to model due to the poorer quality electron density map, and they have a correspondingly higher B-factor. However, Rfree is only a small amount above the average, and still within range for structures of similar resolution that have been deposited in the PDB. We attach the Rfree histogram calculated by PHENIX.REFINE below for the reviewer's reference:

5. Page 8, second paragraph- last sentence: Report refers to Figures 3a and 3b to show a

network of salt bridges and H-bonds. I can't see such interactions shown in that figure. also, this follows by,

Page 9, line 2: can't see T563P mutant shown in supplementary figure 3a

I would list all these key interactions in the figure or in a separate panel. It makes easier for the reader to follow the idea.

6. Page 09- paragraph 1, last sentence: ".....two sets of 6 hydrogen bonds and three salt bridges...". I would suggest authors to list those interactions in a figure or list them in-text.

Author response to point #5 and #6: We now include a schematic diagram depicting the residues and hydrogen bonds that comprise the dimer interface as Figure panel 3f. The figure panel in Supplementary Fig. 3b has been corrected to include the sequence alignment around Thr563. We thank the reviewer for these suggestions.

7. An interesting point is that both SPD mutants, V351 and V347D/V351D still retained the dimer interface. It is unclear what other residues/interactions are driving this dimer formation.

Author response: Based on the reduced dimer formation in the K348A mutant (Supplementary Fig. 5c), we believe that the Lys348-Glu415 interaction contributes to the dimerization. Similarly, other interactions such as Lys345-Glu416 depicted in the diagram in Fig. 3f might promote dimerization.

8. How similar is the structure of mini-SPD to wt-SPD? I would report the RMSD value and any noticeable local structural changes/loop movements here. Also, this can be compared with the AlphaFold structure available and discuss any structural changes.

Author response: We extensively compare the wt-SPD and mini-SPD structures in the section entitled "Structure of monomeric SPD", "Disorder-to-order transition of the oxyanion hole triggered by SPD dimerization", and figure 4. We now report the RMSD value of 0.5 Å in the main text.

The AlphaFold predicted model contains a similar overall fold to that of the wt SPD, with an RMSD of 0.6Å over the Cα atoms. The largest differences are in the β10-β11-α6-α7 protrusion, the backbone conformation of residues in the oxyanion hole are misaligned, and many of the surface-exposed residues are in different conformations.

9. Few missing abbreviations, please use the full term when you use it the first time.

- SV40 and PCNA

Author response: In the revised manuscript, we have spelled out SV40 and PCNA as "simian virus 40" and "proliferating cell nuclear antigen" when they appear the first time.

10. Missing reference on page 4, paragraph 2, line 5: “but FAM111A is one of the few proteases in this family that mainly localizes in the nucleus [REF]”

Author response: We have inserted the following citation, which contains the complete list of human serine proteases in family S1 with P1 specificity and localization in Table S1.

Mehner C., et al. Activity-based protein profiling reveals active serine proteases that drive malignancy of human ovarian clear cell carcinoma. *J Biol Chem* 298, 102146 (2022).

Reviewer #1 (Remarks to the Author):

The manuscript has been revised based on the suggestions of the reviewers and some sections have been done well. However, my concern still remains about the allostery part and changes in the oxyanion hole in the variants as claimed. Without supporting enzymatic data, structural snapshots cannot confirm oxyanion hole dynamics. Furthermore, due to their inability to prove allostery (which was one of the main components of their study), the authors just removed the term from the manuscript which is not only ridiculous but concerning.

Given these limitations, I do not find it suitable for publication in Nature communications.

Reviewer #2 (Remarks to the Author):

The authors addressed all my concerns.

Reviewer #3 (Remarks to the Author):

The authors have addressed essentially all the points I raised in the original report.

Reviewer Comments and Author Responses

Reviewer #1 (Remarks to the Author):

The manuscript has been revised based on the suggestions of the reviewers and some sections have been done well. However, my concern still remains about the allostery part and changes in the oxyanion hole in the variants as claimed. Without supporting enzymatic data, structural snapshots cannot confirm oxyanion hole dynamics. Furthermore, due to their inability to prove allostery (which was one of the main components of their study), the authors just removed the term from the manuscript which is not only ridiculous but concerning.

Given these limitations, I do not find it suitable for publication in Nature communications.

Author response:

We have further refined the description of the structural transition associated with SPD dimerization to remove statements about oxyanion hole dynamics, with the exception of the dynamic movement that occurs during the MD simulations. We note that we defined both a dimeric structure with an oxyanion hole in the correct orientation to support catalysis, and a monomeric structure with oxyanion hole residues that are not positioned correctly, thus identifying active and inactive conformations. We have used site-directed mutagenesis to demonstrate loss of activity attributed to loss of dimerization inactivates protease enzyme activity *in vitro* and loss of function during DNA replication. We are precluded from performing certain kinetic analyses due to the high K_m (above the solubility limit of the peptide substrate) as well as the low K_d for dimer formation. Despite this, the clear “before and after” structures, supporting enzyme assays, and MD simulations to bridge the remaining knowledge gap provide clear evidence for the link between dimerization and SPD protease activity.

Reviewer #2 (Remarks to the Author):

The authors addressed all my concerns.

Reviewer #3 (Remarks to the Author):

The authors have addressed essentially all the points I raised in the original report.